# Rethinking Reasoning in Document Ranking: Why Chain-of-Thought Falls Short

**Xuan Lu**[1,2,3,*], **Haohang Huang**[3,*], **Rui Meng**[†], **Yaohui Jin**[1], **Wenjun Zeng**[2,3], **Xiaoyu Shen**[2,3,‡]

[1]Shanghai Jiao Tong University
[2]Ningbo Key Laboratory of Spatial Intelligence and Digital Derivative
[3]Institute of Digital Twin, Eastern Institute of Technology, Ningbo
`lux1997@sjtu.edu.cn  xyshen@eitech.edu.cn`

## Abstract

Document reranking is a key component in information retrieval (IR), aimed at refining initial retrieval results to improve ranking quality for downstream tasks. Recent studies—motivated by large reasoning models (LRMs)—have begun incorporating explicit chain-of-thought (CoT) reasoning into LLM-based rerankers. However, the effectiveness of such reasoning for ranking tasks remains underexplored. In this work, we present the first systematic study of reasoning in reranking across both logits-based pointwise and listwise settings, under both supervised fine-tuning and reinforcement learning. Using diverse benchmarks, including reasoning-intensive datasets (BRIGHT) and standard IR benchmarks (BEIR), we find that *reasoning-augmented rerankers consistently underperform their direct counterparts that predict rankings without CoT*, despite substantially higher inference costs. Our analysis reveals three core limitations: (i) in pointwise rerankers, reasoning breaks calibration and biases models toward the positive class, raising TPR but lowering TNR, which inflates false positives and degrades ranking in negative-dominant pools; (ii) in listwise rerankers, explicit reasoning improves the fit during training but leads to higher variance and fails to improve performance in both in-domain and out-of-domain evaluations, even when reinforcement learning shortens rationales; and (iii) overall, directly fine-tuned rerankers remain more stable, effective, and robust. These findings challenge the assumption that explicit reasoning is universally beneficial for reranking. We conclude by highlighting future directions, including calibration-aware scoring for pointwise rerankers and the design of concise, targeted reasoning strategies to mitigate overfitting and overthinking in listwise rerankers. [1]

## 1 Introduction

Document reranking is a crucial step in information retrieval (IR), aimed at refining the coarse-grained results produced by first-stage retrieval methods. By reordering candidate documents, reranking improves precision and overall ranking quality, which is essential for downstream applications such as retrieval-augmented generation (RAG) (Lewis et al., 2020; Lu et al., 2025a) and recommendation (Ren et al., 2024). The landscape of reranking is dominated by two primary paradigms: *pointwise* and *listwise*. Pointwise rerankers independently estimate the relevance score of each query–document pair and sort documents accordingly. Since each document is processed in isolation, pointwise rerankers allow parallel computation and efficiency. In contrast, listwise rerankers consider the entire candidate set jointly, asking the model to output a ranked list. While computationally more expensive, listwise rerankers often achieve more accurate rankings by leveraging cross-document interactions and relative comparisons, which is fundamentally easier than assigning a precise relevance score to each document in isolation.

---

[*]Equal contribution
[†]Working at Google Cloud AI Research. ‡ Corresponding author.
[1]We release our code on `https://github.com/EIT-NLP/Direct-Rank`

With the rise of large language models (LLMs), reranking performance has advanced substantially. By combining targeted prompts with task-specific fine-tuning, LLM-based rerankers have achieved state-of-the-art results on diverse benchmarks (Sun et al., 2023a). Recently, large reasoning models (LRMs), such as DeepSeek-R1 (Guo et al., 2025) and OpenAI o1 (Jaech et al., 2024), have further drawn attention. Unlike typical LLMs that directly produce answers, LRMs explicitly decode reasoning chains before providing the final prediction. This process narrows the gap between input and output, smooths token-by-token transitions, and has been shown to improve performance in many tasks. Motivated by these advances, recent studies have sought to extend test-time reasoning to reranking via supervised fine-tuning Weller et al. (2025); Ji et al. (2025); Yang et al. (2025) or using reinforcement learning Zhang et al. (2025); Zhuang et al. (2025); Liu et al. (2025).

Despite these developments, a fundamental question remains unresolved: **Does explicit reasoning truly benefit reranking?** Prior work often assumes that chain-of-thought (CoT) reasoning enhances reranking (Weller et al., 2025; Yang et al., 2025; Zhuang et al., 2025), yet such claims are rarely supported by fair comparisons against non-reasoning baselines (Weller et al., 2025; Yang et al., 2025; Zhuang et al., 2025; Liu et al., 2025). Moreover, emerging evidence suggests that reasoning-augmented rerankers can suffer from *overthinking* and lengthy reasoning chains introduce noise that degrades performance (Jedidi et al., 2025; Fan et al., 2025). However, these analyses are limited in scope, focusing narrowly on pointwise rerankers trained with supervised objectives, and fail to offer a systematic understanding of reasoning's actual role in reranking.

In this work, we present the *first comprehensive and fair study of reasoning in reranking*. To ensure rigor and comparability, we adopt a unified experimental design: all rerankers are trained on the MS MARCO dataset, with reasoning-augmented models using CoT chains generated by DeepSeek-R1. We cover both logits-based *pointwise* and *listwise* rerankers, both *direct-output* and *reasoning-augmented* variants, and both *supervised fine-tuning (SFT)* and *reinforcement learning (RL)* training regimes. This setup eliminates inconsistencies across prior work and allows for a clean, apples-to-apples comparison. We further evaluate models on two complementary benchmarks: BRIGHT, which emphasizes reasoning-intensive queries, and BEIR, a standard suite of retrieval datasets. The scale, diversity, and uniformity of this design ensure that our conclusions are not anecdotal but systematically validated. Our extensive experiments reveal a striking and consistent pattern: under current training and inference setups, *reasoning-based rerankers underperform their direct-output counterparts, even though they incur substantially higher inference costs*. This finding holds across architectures, training strategies, and benchmarks, suggesting that explicit reasoning—which benefits many other LLM tasks—does not translate into gains for reranking. Instead, reasoning introduces calibration errors, overthinking, and poor generalization, ultimately harming ranking quality. Our contributions can be summarized as follows:

- **A rigorous, systematic study.** We conduct the first large-scale, controlled comparison of reasoning vs. direct reranking, covering pointwise and listwise paradigms, SFT and RL training, and both reasoning-intensive and standard IR benchmarks.

- **Clear evidence against reasoning in reranking.** Direct-output rerankers consistently outperform reasoning-augmented variants, despite the latter's substantially higher inference cost.

- **Deeper insights into failure modes.** Our analysis reveals that for pointwise rerankers, reasoning does not improve calibrated relevance prediction; instead, it shifts the error distribution—raising TPR while reducing TNR—which disrupts score calibration and introduces a bias toward false positives. Similarly, for listwise rerankers, reasoning leads to better training fit but increases variance and fails to yield gains on both in-domain and out-of-domain evaluations, even when rationales are shortened via GRPO.

- **Guidance for future research.** Our findings suggest that reranking should prioritize efficient direct scoring rather than reasoning-heavy approaches. Promising directions include calibration-aware scoring for pointwise rerankers and designing concise, targeted reasoning strategies to mitigate overfitting and overthinking in listwise rerankers.

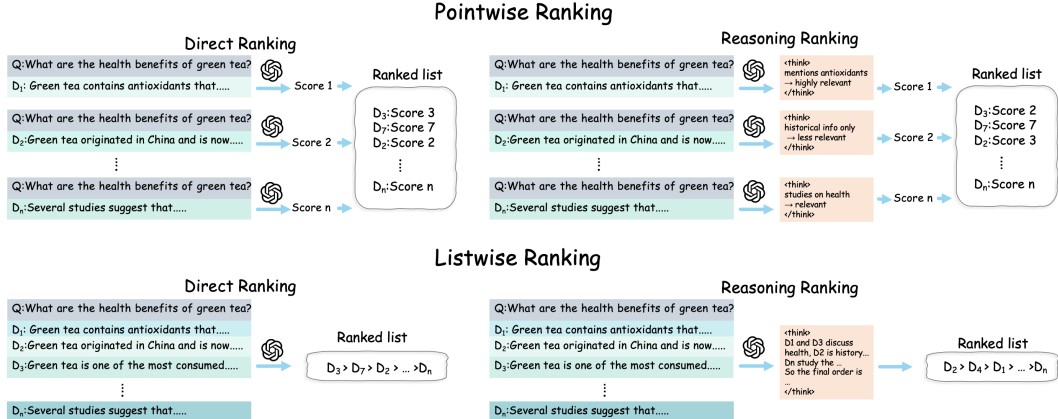

Figure 1: Illustration of Pointwise and Listwise Reranking (Direct vs. Reasoning). In pointwise, each query–document pair is judged independently, with relevance scores computed as the normalized probability of the TRUE token over {TRUE, FALSE} logits. Listwise directly optimizes the ranking order over candidate sets, with or without explicit reasoning.

## 2 PRELIMINARIES

### 2.1 TASK SETUP

We consider the *reranking task* in information retrieval (IR), where the goal is to reorder an initial set of candidate documents so that those most relevant to a query appear at the top. Formally, given a query $q$, a retriever first returns a candidate set

$$C(q) = \{d_1, d_2, \ldots, d_k\}.$$

The reranker then takes $(q, C(q))$ as input and produces an improved ordering of the documents in $C(q)$. This approach reflects the standard two-stage retrieve-and-rerank architecture used in modern IR systems. First, an efficient retriever, optimized for recall, generates an initial, coarsely-ranked list of documents from a large corpus. Then, a reranker, typically a more expressive model optimized for precision, refines this list. The goal of this two-stage process is to maximize user satisfaction by placing the most relevant documents at the very top of the results.

### 2.2 POINTWISE RERANKER

In the pointwise setting, each query–document pair $(q, d_i)$ is evaluated independently. Let $\xi(q, d_i)$ denote the prompt that encodes the pair, and let the answer space be $\mathcal{A} = \{\text{TRUE}, \text{FALSE}\}$, corresponding to tokens $\tau_{\text{TRUE}}$ and $\tau_{\text{FALSE}}$. For candidate $d_i$, the model produces logits $\boldsymbol{\ell}_i \in \mathbb{R}^{|\mathcal{V}|}$, from which a relevance score is derived as the normalized probability of the TRUE token:

$$s_i = \frac{\exp(\boldsymbol{\ell}_i[\tau_{\text{TRUE}}])}{\exp(\boldsymbol{\ell}_i[\tau_{\text{TRUE}}]) + \exp(\boldsymbol{\ell}_i[\tau_{\text{FALSE}}])}.$$

The final ranking is obtained by sorting $\{s_i\}_{i=1}^k$ in descending order.

Optionally, a pointwise reranker can be extended with explicit reasoning: before predicting the binary decision, the model generates an intermediate reasoning trace $z_i$ (e.g., a chain-of-thought),

$$z_i \sim p_\theta\big(z \mid \xi(q, d_i)\big), \qquad s_i = \Pr_\theta\big(a = \text{TRUE} \mid \xi(q, d_i), z_i\big),$$

where $a \in \mathcal{A}$ and the probability is computed from the answer-token distribution conditioned on $z_i$. In practice, multiple traces $\{z_i^{(m)}\}_{m=1}^M$ may be sampled and aggregated (e.g., by averaging or voting). Suppressing $z_i$ recovers the non-reasoning formulation above.

### 2.3 LISTWISE RERANKER

In the listwise setting, the model considers the entire candidate set $C(q)$ jointly. Let $\varphi(q, C(q))$ denote the encoding of the query and its candidate list. The model autoregressively generates a

permutation $\pi = \langle \pi_1, \ldots, \pi_k \rangle$ of indices:

$$\pi \sim p_\theta\big(\pi \mid \varphi(q, C(q))\big), \qquad \pi_j \in \{1, \ldots, k\} \setminus \{\pi_1, \ldots, \pi_{j-1}\}.$$

At inference, one may decode the most likely permutation $\hat{\pi}$ (e.g., via greedy or beam search), or compute ranking scores from partial sequence probabilities. When $k$ exceeds the context window, candidates can be processed in overlapping blocks (e.g., sliding windows), with local rankings merged into a global order.

Optionally, a listwise reranker can be extended with explicit reasoning. In this case, the model first generates a global reasoning trace $Z$ that captures cross-document comparisons:

$$Z \sim p_\theta\big(Z \mid \varphi(q, C(q))\big), \qquad \pi \sim p_\theta\big(\pi \mid \varphi(q, C(q)), Z\big).$$

The trace $Z$ may include pairwise judgments, list-level critiques, or structured deliberation, and can be produced either as a separate stage (generate $Z$ then $\pi$) or interleaved with ranking. Suppressing $Z$ recovers the standard listwise formulation described above.

## 3  EVALUATING THE IMPACT OF REASONING ON RERANKING

### 3.1  MODEL VARIANTS

We design four LLM-based rerankers, each corresponding to a different combination of *pointwise vs. listwise* and *reasoning vs. non-reasoning* paradigms:

- **Direct-Point** (Non-Reasoning Pointwise): the model directly outputs a binary relevance decision (TRUE/FALSE). We take the logits of the answer token and transform them into a probability score, which is used for ranking.

- **Reason-Point** (Reasoning Pointwise): the model first generates a reasoning trace describing why the document may or may not be relevant, and then produces the final binary decision. The relevance score is computed from the logits at the answer token position.

- **Direct-List** (Non-Reasoning Listwise): the model takes the entire candidate list as input and directly generates a permutation as the output ranking, e.g., $[3] > [5] > [4] > \cdots$.

- **Reason-List** (Reasoning Listwise): the model first generates a reasoning sequence that compares and analyzes candidates, and then outputs the final ranking sequence.

### 3.2  TRAINING DETAILS

**Backbone Models**  We adopt the Qwen3 series as the backbone for our rerankers, specifically Qwen3-4B and Qwen3-8B. This choice is consistent with the prevailing practice in the reranking community[2]. All training experiments are conducted on two NVIDIA A100 (80 GB) GPUs.

**Pointwise Rerankers**  We study two pointwise variants: *Direct-Point* and *Reason-Point*. Both models are trained on the RANK1 corpus (Weller et al., 2025) derived from MS MARCO, comprising $\sim 386$k query–passage pairs annotated by DeepSeek-R1 with a chain-of-thought rationale and a binary answer (TRUE/FALSE). For *Reason-Point*, we perform supervised fine-tuning on quadruples $\langle$query, passage, rationale, answer$\rangle$. For *Direct-Point*, we ablate the rationale and fine-tuning on $\langle$query, passage, answer$\rangle$, training the model to emit a single token in $\{$TRUE, FALSE$\}$. Both variants minimize cross-entropy loss. We employ the LLaMA-Factory[3] framework for supervised fine-tuning. All models are trained using LoRA with rank 32 and $\alpha = 64$, a learning rate of $1 \times 10^{-4}$, and cross-entropy loss. Both DIRECT-POINT and REASON-POINT rerankers are trained for one epoch. At inference time, we compute the relevance score used for ranking as the probability assigned to TRUE via a two-way softmax over the logits of $\{$TRUE, FALSE$\}$; ties are broken by the logit margin. Example prompts and data instances for both settings are provided in the Appendix C.1.

---

[2]A complete list of backbone models used in both our work and prior reasoning-enhanced rerankers is provided in Appendix A.

[3]https://github.com/hiyouga/LLaMA-Factory

**Listwise Rerankers** We train listwise rerankers on the REASONRANK training corpus (Liu et al., 2025), which contains ∼13k query–candidate sets primarily derived from MS MARCO and related benchmarks, split evenly between SFT and GRPO (approximately 6.7k each). Each instance includes a query, a candidate set, a rationale produced with DeepSeek-R1, and a gold ranking order. We consider two variants: *Direct-List*, which generates an ordering directly from the query and candidate set, and *Reason-List*, which is prompted to first generate own rationale and then produce the final ordering.

We adopt two-stage training for traning *Direct-List* and *Reason-List* Stage 1 performs supervised fine-tuning (SFT) to teach the model to output ranking sequences. To encourage structurally valid outputs, prompts require the model to produce a reasoning segment demarcated by `<think>···</think>` and a final ranking in `<answer>[·] > [·] > ···</answer>` format. Stage 2 refines the SFT model with Group Relative Policy Optimization (GRPO) (Guo et al., 2025). We follow the setting in ReasonRank (Liu et al., 2025), using a composite *multi-view* ranking reward that reflects position sensitivity, coverage, and list similarity:

Let $y^{\text{list}}$ denote the predicted ranking sequence and $y'$ the gold ranking. We combine three signals:

$$R_m = \text{NDCG@10}(y^{\text{list}}, y') \ + \ \phi \cdot \text{Recall@10}(y^{\text{list}}, y') \ + \ \gamma \cdot \text{RBO}(y^{\text{list}}, y'), \quad (1)$$

where $\phi, \gamma$ weight coverage and overlap. Rank-Biased Overlap (RBO) (Webber et al., 2010) emphasizes top ranks and is computed as

$$\text{RBO}(y^{\text{list}}, y') \ = \ (1-p) \sum_{d=1}^{|y^{\text{list}}|} p^{d-1} \, \frac{\left| y^{\text{list}}_{1:d} \cap y'_{1:d} \right|}{d}, \quad (2)$$

with persistence parameter $p \in (0,1)$ and $y_{1:d}$ the top-$d$ prefix. Following REASONRANK, we gate $R_m$ with simple format validators to stabilize learning:

$$R \ = \ \begin{cases} R_m, & \text{both output and answer formats are valid,} \\ 0, & \text{only the output format is valid,} \\ -1, & \text{otherwise,} \end{cases} \quad (3)$$

where the *output-format* check requires the presence of `<think>` and `<answer>` tags, and the *answer-format* check verifies that the `<answer>` contains a canonical listwise ordering.

With the gated multi-view reward in Eqs. (1)–(2), we refine the SFT policy via GRPO (Guo et al., 2025). Given input $x$, a group of samples $\mathcal{G} = \{y_i\}$ is drawn, sequence rewards $R(x, y_i)$ are converted to token-level advantages $\widehat{A}_{i,t}$, and the policy is updated by the clipped objective:

$$\mathcal{J}_{\text{GRPO}}(\theta) = -\frac{1}{|\mathcal{G}|} \sum_{i,t} \min\left(r_{i,t}(\theta)\widehat{A}_{i,t}, \ \text{clip}(r_{i,t}(\theta), 1-\epsilon, 1+\epsilon)\widehat{A}_{i,t}\right) \ - \ \beta \, D_{\text{KL}}(\pi_\theta \, \| \, \pi_{\text{ref}}), \quad (4)$$

where $r_{i,t}(\theta)$ is the importance ratio and $\pi_{\text{ref}}$ the SFT reference. We implement GRPO with `Verl`[4] for one epoch on the ReasonRank data, producing *Direct-List* and *Reason-List* models.

### 3.3 EXPERIMENTAL SETUP

All experiments are conducted on two NVIDIA A100 (80 GB) GPUs. Rerankers operate on a fixed first-stage candidate pool of $k{=}100$ passages per query built with BM25. Unified system prompts are used across all datasets, as shown in Appendix C. Dataset-level instructions strictly follow the original Rank1 templates (Weller et al., 2025).

**Baselines** We compare our four proposed rerankers—*Direct-Point*, *Reason-Point*, *Direct-List*, and *Reason-List*—including ablations across training stages (SFT only vs. SFT+GRPO). In addition, we compare our models against state-of-the-art *reasoning-enhanced* LLM rerankers from prior work: Pointwise: Rank1-7B, Rank1-14B (Weller et al., 2025), TF-Rank-4B, TF-Rank-8B (Fan et al., 2025); Listwise: Rank-R1-7B, Rank-R1-14B (Zhuang et al., 2025), REARank-7B (Zhang et al., 2025), and ReasonRank-7B (Liu et al., 2025).

---

[4] `https://github.com/volcengine/verl`

**Benchmarks and Metrics**   We evaluate on two retrieval benchmarks: BRIGHT, a reasoning-intensive IR suite spanning diverse domains, and BEIR, a standard heterogeneous IR benchmark. Following common practice, the primary metric is NDCG@10, which captures both relevance and position sensitivity on the top-10 results.

### 3.4   MAIN RESULTS

**Reasoning Does Not Improve Reranking Performance**   As shown in Tables 1 and 2, a consistent and repeatable pattern emerges across *all* training settings (SFT and SFT+GRPO), model sizes (4B/8B), and benchmarks (BRIGHT and BEIR): ***direct* rerankers consistently outperform their *reasoning*-augmented counterparts**. For pointwise rerankers on BRIGHT, *Direct-Point-4B* exceeds *Reason-Point-4B* by $\Delta N@10 = +9.0$, and *Direct-Point-8B* by $+6.1$ on the original query, while the advantage remains $+4$–$6$ points on the gpt4_reason query split.[5]; on BEIR, the corresponding gaps are $+5.3$ and $+4.3$. For listwise rerankers, the advantage is smaller but remains stable: on BRIGHT, *Direct-List* achieves $+0.4$–$+1.7$ higher $N@10$ than *Reason-List*, while on BEIR the margin ranges from $+0.3$ to $+1.9$, consistently observed under both SFT and SFT+GRPO. Moreover, GRPO provides further improvements for listwise rerankers compared to SFT alone, yet the superiority of direct reranking persists regardless of training strategy. These results reveal one clear trend: under current logits-based pointwise and generative listwise reranking setups, we observe that explicit reasoning does *not* improve ranking performance.

**Comparison Results on BRIGHT and BEIR.**   Tables 1 and 2 report the performance of our rerankers compared to reasoning-enhanced baselines on BRIGHT and BEIR, respectively. On original query split of BRIGHT, our *Direct-List-8B* achieves the best overall score with $N@10 = 27.1$, followed closely by *Direct-Point-8B* at $26.8$, both outperforming reasoning-based baselines such as *ReasonRank-7B* ($26.4$), *TFRank-8B* ($22.6$), and *Rank-R1-14B* ($20.5$). On the gpt4_reason split of BRIGHT, *Direct-List-4B* and *Direct-List-8B* (both $N@10 = 35.3$) likewise surpass all Reason-List baselines. On BEIR, the strongest model is *Direct-Point-4B*, which obtains $N@10 = 45.4$, surpassing larger reasoning-enhanced listwise rerankers such as *Rank-R1-14B* ($43.8$) and *ReasonRank-7B* ($41.7$). These results demonstrate that our *direct* rerankers not only outperform existing reasoning-based counterparts, but also highlight that explicit reasoning is unnecessary for achieving state-of-the-art effectiveness in LLM-based reranking.

## 4   ANALYSIS

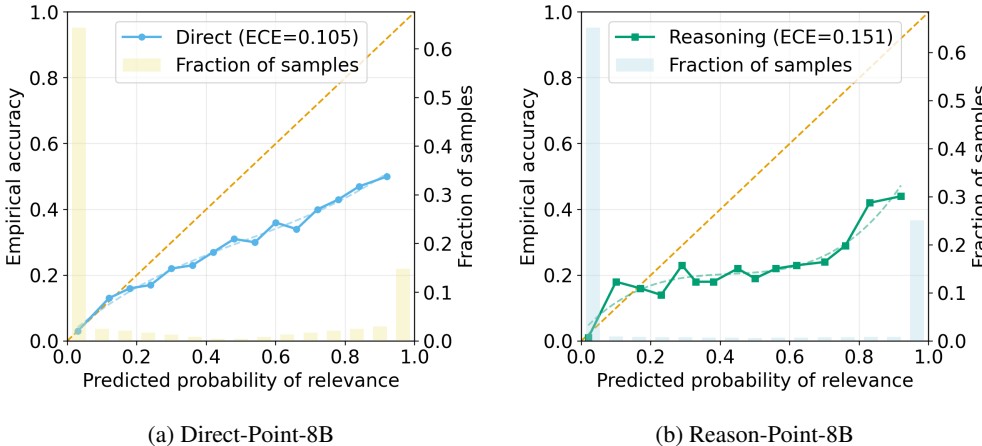

(a) Direct-Point-8B                    (b) Reason-Point-8B

Figure 2: Calibration curves of pointwise rerankers: predicted probabilities vs. empirical accuracies.

---

[5]Complete results for the gpt4_reason split are reported in Appendix B.

Table 1: Performance comparison on BRIGHT across different reranker variants. We report results for Direct-Point, Reason-Point, Direct-List, and Reason-List under both SFT and GRPO training, together with representative pointwise and listwise baselines.

| Model | Training | StackExchange | | | | | | | Coding | | Theorem-based | | | Avg. |
|---|---|---|---|---|---|---|---|---|---|---|---|---|---|---|
| | | Bio. | Earth. | Econ. | Psy. | Rob. | Stack. | Sus. | Leet. | Pony | AoPS | TheoQ. | TheoT. | |
| **Pointwise** | | | | | | | | | | | | | | |
| BM25 | / | 18.9 | 27.2 | 14.9 | 12.5 | 13.6 | 18.4 | 15.0 | 7.9 | 24.4 | 6.2 | 4.9 | 10.4 | 14.5 |
| Rank1-7B | SFT | 31.4 | 36.7 | 18.3 | 25.4 | 13.8 | 17.6 | 24.8 | 16.7 | 9.5 | 6.1 | 9.5 | 11.6 | 18.5 |
| Rank1-14B | SFT | 29.6 | 34.8 | 17.2 | 24.3 | 18.6 | 16.2 | 24.5 | 17.5 | 14.4 | 5.5 | 9.2 | 10.7 | 18.5 |
| TFRank-4B | SFT+GRPO | 33.2 | 45.9 | 17.6 | 29.5 | 21.0 | 20.9 | 18.3 | 25.0 | 9.1 | 9.5 | 9.8 | 7.3 | 20.6 |
| TFRank-8B | SFT+GRPO | 33.7 | 46.2 | 23.7 | 26.0 | 24.1 | 20.1 | 23.6 | 28.8 | 12.5 | 10.8 | 11.4 | 9.7 | 22.6 |
| **Reason-Point-4B** | SFT | 23.6 | 29.0 | 15.0 | 23.7 | 16.7 | 12.2 | 18.3 | 18.4 | 12.4 | 8.9 | 11.0 | 9.4 | 16.5 |
| **Direct-Point-4B** | SFT | 34.9 | 45.1 | 23.3 | 31.8 | 26.6 | 23.6 | 30.7 | 18.5 | 35.4 | 7.2 | 13.6 | 15.2 | 25.5 |
| **Reason-Point-8B** | SFT | 24.9 | 34.6 | 17.5 | 26.2 | 25.9 | 22.4 | 19.7 | 11.9 | 36.6 | 9.3 | 6.5 | 12.6 | 20.7 |
| **Direct-Point-8B** | SFT | 33.9 | 46.4 | 24.6 | 31.6 | 25.8 | 25.9 | 32.0 | 25.3 | 35.5 | 12.0 | 13.5 | 15.2 | **26.8** |
| **Listwise** | | | | | | | | | | | | | | |
| Rank-R1-7B | GRPO | 26.0 | 28.5 | 17.2 | 24.2 | 19.1 | 10.4 | 24.2 | 19.8 | 4.3 | 4.3 | 8.3 | 10.9 | 16.4 |
| Rank-R1-14B | GRPO | 31.2 | 38.5 | 21.2 | 26.4 | 22.6 | 18.9 | 27.5 | 20.2 | 9.2 | 9.7 | 9.2 | 11.9 | 20.5 |
| REARANK-7B | GRPO | 23.4 | 27.4 | 18.5 | 24.2 | 17.4 | 16.3 | 25.1 | 27.0 | 8.0 | 7.4 | 7.9 | 9.5 | 17.7 |
| ReasonRank-7B | SFT+GRPO | 36.3 | 44.2 | 24.8 | 31.7 | 30.7 | 24.9 | 32.8 | 28.7 | 17.5 | 12.0 | 18.5 | 14.0 | 26.4 |
| **Reason-List-4B** | SFT | 30.7 | 37.3 | 18.7 | 27.7 | 27.9 | 19.8 | 28.5 | 28.1 | 13.7 | 9.1 | 13.9 | 13.3 | 22.4 |
| **Direct-List-4B** | SFT | 32.7 | 38.6 | 20.0 | 28.4 | 28.6 | 20.5 | 31.2 | 30.9 | 15.1 | 10.4 | 17.8 | 15.6 | 24.1 |
| **Reason-List-8B** | SFT | 31.9 | 39.6 | 22.4 | 29.0 | 29.9 | 23.4 | 34.5 | 26.8 | 18.9 | 9.7 | 15.6 | 12.1 | 24.5 |
| **Direct-List-8B** | SFT | 32.6 | 38.4 | 21.3 | 28.9 | 31.9 | 22.6 | 31.8 | 28.9 | 16.9 | 11.1 | 18.5 | 15.4 | 24.9 |
| **Reason-List-4B** | SFT+GRPO | 33.6 | 40.8 | 21.6 | 28.0 | 33.3 | 26.0 | 29.3 | 31.0 | 13.3 | 11.4 | 16.5 | 15.4 | 25.0 |
| **Direct-List-4B** | SFT+GRPO | 33.8 | 41.5 | 23.4 | 29.3 | 34.0 | 23.9 | 34.2 | 33.4 | 13.7 | 11.9 | 17.1 | 14.6 | 25.9 |
| **Reason-List-8B** | SFT+GRPO | 32.1 | 40.3 | 26.7 | 32.1 | 30.0 | 25.5 | 33.8 | 28.8 | 19.4 | 9.8 | 18.0 | 14.0 | 25.9 |
| **Direct-List-8B** | SFT+GRPO | 35.2 | 42.7 | 23.1 | 30.6 | 34.0 | 27.6 | 33.9 | 29.2 | 22.9 | 12.1 | 17.9 | 15.8 | **27.1** |

## 4.1 Pointwise: Calibration Failure and True Bias with Reasoning

**Reasoning breaks calibration of confidence and accuracy** Calibration assesses whether predicted probabilities match the true likelihood of relevance. In pointwise rerankers, the score assigned to a candidate is interpreted as the model's confidence that it is relevant. A perfectly calibrated model satisfies, for example, that predictions of 0.9 correspond to roughly 90% truly relevant items; in reliability diagrams this appears as points along the diagonal $y=x$. To quantify deviations from perfect calibration, we use the *Expected Calibration Error* (ECE) (Guo et al., 2017), the weighted discrepancy between predicted confidence and empirical accuracy across $M$ bins: $\text{ECE} = \sum_{m=1}^{M} \frac{|B_m|}{N} \left| \text{acc}(B_m) - \text{conf}(B_m) \right|$, where $B_m$ is the set of samples in bin $m$, $|B_m|$ its size, $N$ the total number of samples, $\text{acc}(B_m)$ the empirical accuracy, and $\text{conf}(B_m)$ the average predicted probability in that bin. Smaller values indicate better calibration. Notably, we observe a pronounced polarization in the logits: most predictions cluster near 0 or 1, reflecting overconfident decision boundaries. As shown in Figure 2, based on results from the BEIR, the direct pointwise reranker—though not perfect—maintains a clear monotonic relationship between confidence and accuracy (ECE = 0.105). By contrast, the reasoning-enhanced reranker exhibits systematic overconfidence with larger departures from the diagonal (ECE = 0.151), indicating that *adding reasoning breaks confidence calibration* in the pointwise setting. This miscalibration helps explain the observed degradation in ranking quality (e.g., lower NDCG).

**Reasoning increases "True" proclivity** Beyond aggregate calibration, class-conditional analysis reveals a consistent shift toward predicting the positive class. The training data have a positive:negative ratio of approximately 1:2. To match this prior, we construct evaluation pools with 100 positives and 200 negatives per query—both in-domain (MS MARCO DL19/DL20) and out-of-domain (BRIGHT–Biology)—and also include the teacher model (DeepSeek-R1). Letting the reranker judge relevance and decode the answer token, we report class-conditional performance in Table 3 using standard notation: TPR (true positive rate; recall on positives) and TNR (true negative rate; specificity = $1-$FPR). Under this matched prior, *Reason* models tend to achieve higher macro binary accuracy (the mean of TPR and TNR) than *Direct* models; however, the gains consistently

Table 2: Performance comparison on BEIR across different reranker variants. We report results for Direct-Point, Reason-Point, Direct-List, and Reason-List under both SFT and GRPO training, together with representative pointwise and listwise baselines

| Model | Training | ArguA | ClimF | DBP | FiQA | NFCorp | SciDoc | SciFact | Touche | TrecC | Avg. |
|---|---|---|---|---|---|---|---|---|---|---|---|
| | | | | | Pointwise | | | | | | |
| BM25 | / | 39.7 | 16.5 | 31.8 | 23.6 | 33.8 | 14.9 | 67.9 | 44.2 | 59.5 | 36.9 |
| Rank1-7B | SFT | 26.4 | 16.2 | 37.7 | 38.4 | 37.9 | 16.5 | 76.1 | 24.5 | 79.5 | 39.2 |
| Rank1-14B | SFT | 32.2 | 15.6 | 34.2 | 36.6 | 35.1 | 16.6 | 73.8 | 25.9 | 78.0 | 38.7 |
| TF-Rank-4B | SFT+GRPO | 37.2 | 19.7 | 37.9 | 36.2 | 38.3 | 18.3 | 76.6 | 37.6 | 80.5 | 42.5 |
| TF-Rank-8B | SFT+GRPO | 36.5 | 21.7 | 37.0 | 38.0 | 38.0 | 17.9 | 74.6 | 35.0 | 80.0 | 42.1 |
| **Reason-Point-4B** | SFT | 38.1 | 16.5 | 39.1 | 36.0 | 30.8 | 16.8 | 74.9 | 27.1 | 81.3 | 40.1 |
| **Direct-Point-4B** | SFT | 57.5 | 19.6 | 43.4 | 42.4 | 35.9 | 18.6 | 77.4 | 30.6 | 83.5 | **45.4** |
| **Reason-Point-8B** | SFT | 40.1 | 13.8 | 37.5 | 35.6 | 32.5 | 18.2 | 74.1 | 21.6 | 80.3 | 39.3 |
| **Direct-Point-8B** | SFT | 58.6 | 15.7 | 42.7 | 43.3 | 36.2 | 16.8 | 75.3 | 22.3 | 82.7 | 43.7 |
| | | | | | Listwise | | | | | | |
| Rank-R1-7B | GRPO | 37.0 | 24.1 | 43.2 | 40.1 | 36.2 | 18.8 | 76.1 | 33.0 | 82.6 | 43.5 |
| Rank-R1-14B | GRPO | 34.4 | 24.2 | 44.0 | 43.0 | 37.9 | 19.7 | 77.5 | 29.6 | 83.9 | **43.8** |
| REARANK-7B | GRPO | 35.6 | 20.6 | 43.5 | 35.8 | 37.9 | 19.2 | 71.9 | 40.2 | 80.1 | 42.8 |
| ReasonRank-7B | SFT+GRPO | 33.3 | 20.0 | 44.7 | 38.2 | 36.6 | 19.7 | 72.8 | 30.4 | 79.6 | 41.7 |
| **Reason-List-4B** | SFT | 39.3 | 13.8 | 37.7 | 32.9 | 30.2 | 16.0 | 69.1 | 24.6 | 79.5 | 38.1 |
| **Direct-List-4B** | SFT | 41.7 | 14.1 | 36.5 | 37.0 | 33.5 | 16.4 | 72.4 | 22.9 | 78.1 | 39.2 |
| **Reason-List-8B** | SFT | 32.8 | 16.2 | 42.2 | 36.6 | 36.0 | 18.1 | 69.7 | 27.1 | 79.2 | 39.8 |
| **Direct-List-8B** | SFT | 28.9 | 16.4 | 42.3 | 37.7 | 35.9 | 18.9 | 73.6 | 27.0 | 80.1 | 40.1 |
| **Reason-List-4B** | SFT+GRPO | 30.8 | 15.7 | 43.3 | 36.1 | 36.5 | 18.2 | 73.1 | 26.4 | 78.0 | 39.8 |
| **Direct-List-4B** | SFT+GRPO | 36.7 | 19.4 | 43.8 | 34.4 | 36.4 | 18.2 | 69.8 | 29.4 | 77.7 | 40.6 |
| **Reason-List-8B** | SFT+GRPO | 28.6 | 19.9 | 43.5 | 35.5 | 37.2 | 18.8 | 72.5 | 24.9 | 77.8 | 39.9 |
| **Direct-List-8B** | SFT+GRPO | 36.4 | 19.4 | 45.1 | 38.2 | 36.9 | 18.7 | 71.0 | 31.6 | 78.7 | 41.8 |

Table 3: Class-conditional performance on pointwise rerankers. We report $TPR$ (%) and $TNR$ (%).

| Model | Biology | | MS MARCO | | Avg. |
|---|---|---|---|---|---|
| | TPR | TNR | TPR | TNR | |
| DeepSeek-R1 | **52.4** | 96.1 | **40.8** | 85.1 | **68.6** |
| Reason-Point-4B | 43.7 | 91.3 | 38.7 | 79.4 | 63.3 |
| Direct-Point-4B | 34.0 | 93.2 | 30.7 | 85.7 | 60.9 |
| Reason-Point-8B | 50.5 | 98.1 | 35.9 | 85.5 | 67.5 |
| Direct-Point-8B | 31.1 | **100.0** | 25.5 | **94.2** | 62.7 |

Table 4: Listwise (GRPO) performance on MS MARCO (NDCG@10).

| Model | MS MARCO | |
|---|---|---|
| | DL19 | DL20 |
| Direct-List-4B | **73.77** | 68.97 |
| Reason-List-4B | 70.76 | 68.71 |
| Direct-List-8B | 73.00 | **71.38** |
| Reason-List-8B | 72.60 | 69.81 |

arise from *higher TPR* coupled with *lower TNR* (i.e., higher FPR). In reranking regimes where negatives dominate, this combination is detrimental: elevated FPR promotes non-relevant documents into the head of the ranked list and, together with the calibration failure above, prevents the binary accuracy gains from translating into improved ranking metrics.

## 4.2 LISTWISE: REASONING IMPROVES TRAINING FIT BUT HURTS GENERALIZATION

**Reasoning boosts training fit but hurts generalization** Unlike pointwise rerankers that assign scores to query–document pairs and then sort, listwise objectives *directly optimize the permutation* of a candidate set. We therefore ask whether exposing chain-of-thought (CoT) helps under listwise training. We evaluate four 8B variants on a training split of 100 instances: *Direct-List_SFT* vs. *Reason-List_SFT* and *Direct-List_GRPO* vs. *Reason-List_GRPO*. As shown in Fig. 3, reasoning attains higher mean NDCG@10 on the training split but with markedly larger dispersion: *Reason-List_SFT* 82.57 ± 3.2 vs. *Direct-List_SFT* 80.41 ± 2.1 (Δ= + 2.16), and *Reason-List_GRPO* 87.55 ± 2.7 vs. *Direct-List_GRPO*

86.93 $\pm$ 1.6 ($\Delta= + 0.62$). These patterns indicate that CoT can better fit the target permutations encountered during training, while simultaneously introducing instance-level instability.

On the in-domain MS MARCO Dev sets (DL19/20), *Direct-List* consistently outperforms *Reason-List* across both 4B and 8B backbones (Table 4). Concretely, *Direct-List-4B* surpasses its reasoning counterpart by $+3.01$ (73.77 vs. 70.76) on DL19 and $+0.26$ (68.97 vs. 68.71) on DL20; *Direct-List-8B* leads by $+0.40$ (73.00 vs. 72.60) on DL19 and $+1.57$ (71.38 vs. 69.81) on DL20. Thus, the training-split advantage of *Reason-List* does *not* translate to stronger in-domain performance. The same trend holds on BRIGHT and BEIR: reasoning-based listwise models lag behind their direct counterparts, reinforcing that CoT's gains on the training split reflect improved *in-sample* fitting rather than genuine *out-of-domain* generalization.

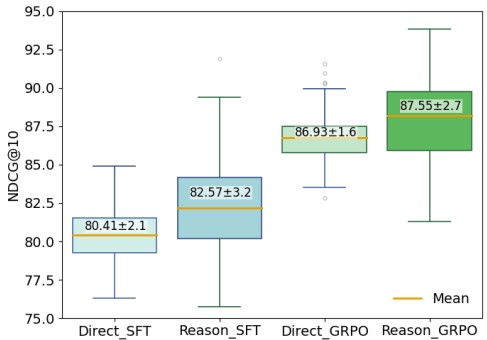

Figure 3: Training-split listwise performance of four 8B variants. Reasoning improves mean NDCG@10 but increases variance.

**GRPO improves performance and reduces overthinking.** As shown in Table 1, Table 2, and Fig. 3, GRPO yields substantial performance improvements over SFT training. At the same time, GRPO markedly shortens the rationales produced by reasoning models. On the training split (Fig. 3), the average rationale length decreases from 397.7 tokens in *Reason-List_SFT* to 172.3 in *Reason-List_GRPO*, reducing inference cost and mitigating extreme "overthinking," yet achieving higher NDCG scores. This finding suggests that excessively long CoT rationales are not a prerequisite for producing effective ranking orders. The compression effect follows directly from the GRPO reward design (Eq. 3), which incentivizes only output format validity and ranking quality rather than verbose reasoning. Although GRPO enhances stability and efficiency, *Direct-List* models still achieve stronger generalization on MS MARCO DL19/20 and on BRIGHT/BEIR (Table 4), implying that shorter CoT mitigates overthinking but does not substitute for direct optimization of permutations. These observations point to a broader research direction: future work should explore how to design *concise and targeted* reasoning strategies that balance interpretability, stability, and generalization, while avoiding overfitting and reliance on lengthy CoT outputs.

## 4.3 Implications for Future Research

Our findings show that explicit reasoning does not inherently lead to performance gains in reranking, and therefore researchers should not assume that longer or more elaborate reasoning will universally improve ranking quality. Instead, the results point to two concrete directions for future work.

For **logits-based pointwise rerankers**, the performance degradation is primarily driven by score miscalibration rather than insufficient reasoning capacity. This suggests that future progress is more likely to come from *calibration-aware training objectives* that preserve score monotonicity, rather than increasing the depth or length of reasoning traces. Beyond calibration, another promising direction is to explore *token-based scoring and ranking mechanisms* (Fan et al., 2025), which operate at the token level rather than relying solely on a scalar logit. Such methods can produce more stable and fine-grained relevance signals, and therefore may complement calibration-oriented approaches in improving pointwise reranking.

For **listwise reranking**, we observe that the training data may contain unnecessarily long chain-of-thought traces, which increase inference cost without translating into better ranking quality. This suggests two promising directions: (i) developing *more concise or adaptive reasoning strategies* to reduce overthinking(Chen et al., 2025), and (ii) designing *reward formulations directly aligned with ranking metrics*, rather than relying on generic reasoning supervision.

Overall, direct models remain more stable and effective across both in-domain and OOD settings, highlighting *calibration* (for pointwise rerankers) and *objective alignment* (for listwise training) as the key bottlenecks for future research, rather than reasoning capacity itself.

## 5 RELATED WORKS

**LLMs for Ranking**  The use of Large Language Models (LLMs) for ranking has emerged as a dominant paradigm in information retrieval, with methods that can be broadly classified into two primary approaches: *pointwise* and *listwise* (Qin et al., 2024; Lu et al., 2025b). Pointwise rerankers evaluate the relevance of each query-document pair in isolation. This is typically achieved by framing the task as a classification problem, where the model computes a relevance score from the output logits of binary tokens like "true" or "false". This approach, exemplified by influential models such as MonoT5 (Nogueira et al., 2020), MonoBERT (Nogueira et al., 2019), and RankLLaMA (Ma et al., 2024), benefits from simplicity and computational efficiency, as each document can be scored independently. In contrast, the listwise paradigm is built on the principle of relative comparison, where multiple candidate documents are considered jointly to determine their final order. This category encompasses several implementation styles. The most direct application is generative listwise ranking, where models like RankGPT (Sun et al., 2023b), RankVicuna (Pradeep et al., 2023a), and RankZephyr (Pradeep et al., 2023b) leverage their generative capabilities to output a fully sorted list of documents. This broader paradigm also includes pairwise methods, which learn relative preferences by predicting the more relevant document from a pair (Qin et al., 2024), and setwise approaches that operate on a group of candidates, for example by identifying the single most relevant document within the set (Zhuang et al., 2024). To manage the long input sequences inherent to this approach, many listwise methods employ strategies like sliding windows or hierarchical ranking to improve efficiency (Sharifymoghaddam et al., 2025).

**LRMs for Ranking**  Inspired by the success of Large Reasoning Models (LRMs), a recent line of research has focused on incorporating explicit reasoning into the reranking process to handle complex queries and improve interpretability. This effort has largely followed two main technical strategies: supervised fine-tuning with distillation and reinforcement learning. The first strategy involves *Supervised Fine-Tuning (SFT) and distillation*, where reasoning capabilities are transferred from powerful teacher models (e.g., GPT-4, DeepSeek-R1) to smaller, more efficient rerankers. For instance, ReasoningRank (Ji et al., 2025) and Rank1 (Weller et al., 2025) distill pairwise or listwise comparative rationales into models such as the LLaMA-3 and Qwen2.5 series. This approach has also been extended to generative listwise settings, where reasoning tokens are directly integrated into the ranking sequence (Yang et al., 2025). A complementary strategy employs *Reinforcement Learning* (RL) to further refine these reasoning-aware models. Works like Rank-R1 (Zhuang et al., 2025) and REARANK (Zhang et al., 2025) use RL to directly optimize for ranking metrics, while others such as TFRank (Fan et al., 2025) and ReasonRank (Liu et al., 2025) adopt a hybrid two-stage approach combining SFT with subsequent RL fine-tuning. Beyond these dominant paradigms, other methods have explored more structured formulations, such as modeling reranking as a decision process to improve robustness (Lee et al., 2025; Niu et al., 2024). Despite these advances, the core premise that explicit reasoning is beneficial is being called into question. Recent findings suggest that for pointwise rerankers, the addition of reasoning can be detrimental, leading to issues like overthinking (Jedidi et al., 2025; Fan et al., 2025). This emerging evidence highlights that the utility of reasoning in reranking is far from settled, motivating the systematic investigation in our work.

## 6 CONCLUSION

In this work, we systematically examined the role of explicit reasoning in document reranking across pointwise and listwise paradigms with SFT and GRPO training. Our findings are threefold: (i) in pointwise rerankers, reasoning breaks calibration, yielding overconfident scores and degraded ranking despite modest gains in binary accuracy; (ii) reasoning biases models toward the positive class, raising TPR but reducing TNR, which is harmful in negative-dominant candidate pools; (iii) in listwise rerankers, reasoning improves in-domain fit but increases variance and fails to generalize out-of-domain, even when GRPO shortens rationales. Overall, direct models remain more stable and effective, pointing to the need for calibration-aware objectives in pointwise rerankers and more concise reasoning strategies in listwise rerankers to improve generalization.

## ETHICS STATEMENT

This work does not involve human subjects, personal data, or sensitive information, and therefore raises no direct ethical concerns. Our research focuses on reranking methods for information retrieval benchmarks, which do not include personally identifiable information or sensitive content. We follow the ICLR Code of Ethics and ensure that our methodology and results are presented with transparency and fairness.

## REPRODUCIBILITY STATEMENT

We are committed to ensuring the reproducibility of our results. All datasets used in this paper are publicly available, and detailed descriptions of preprocessing steps are provided in the appendix. In addition, we will release our source code, trained models, and experiment configurations upon publication to facilitate replication of our experiments. This includes scripts for data preprocessing, training, and evaluation, ensuring that other researchers can reproduce our findings.

### ACKNOWLEDGEMENTS

We thank EIT and IDT High Performance Computing Center for providing computational resources for this project. This work was supported by the 2035 Key Research and Development Program of Ningbo City under Grant No.2025Z034.

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

## A  BACKBONES OF RERANKERS

Table 5 summarizes the backbones and training strategies of both existing reasoning-enhanced rerankers and our proposed models. We observe that the **Qwen** family has become the mainstream backbone for LLM-based reranking. Our proposed *Direct Rerankers* and *Reason Rerankers* are built upon Qwen3-4B and Qwen3-8B, ensuring a fair comparison with recent reasoning-enhanced baselines while highlighting the impact of reasoning versus direct decision-making.

Table 5: Overview of baseline and proposed rerankers.

| Model | Training | Backbone | Type |
|---|---|---|---|
| BM25 | / | / | Pointwise |
| Rank1-7B | SFT | Qwen2.5-7B | Pointwise |
| Rank1-14B | SFT | Qwen2.5-14B | Pointwise |
| TFRank-4B | SFT + GRPO | Qwen3-4B | Pointwise |
| TFRank-8B | SFT + GRPO | Qwen3-8B | Pointwise |
| REARANK-7B | GRPO | Qwen2.5-7B | Listwise |
| Rank-R1-7B | GRPO | Qwen2.5-7B | Listwise |
| Rank-R1-14B | GRPO | Qwen2.5-14B | Listwise |
| ReasonRank-7B | SFT + GRPO | Qwen2.5-7B | Listwise |
| Direct-Point-4B | SFT | Qwen3-4B | Pointwise |
| Direct-Point-8B | SFT | Qwen3-8B | Pointwise |
| Reason-Point-4B | SFT | Qwen3-4B | Pointwise |
| Reason-Point-8B | SFT | Qwen3-8B | Pointwise |
| Direct-List-4B | SFT + GRPO | Qwen3-4B | Listwise |
| Direct-List-8B | SFT + GRPO | Qwen3-8B | Listwise |
| Reason-List-4B | SFT + GRPO | Qwen3-4B | Listwise |
| Reason-List-8B | SFT + GRPO | Qwen3-8B | Listwise |

## B  EXPERIMENTAL RESULTS FOR gpt4_reason QUERY

Table 6 reports the performance of Reason rerankers and Direct rerankers under different training stages on BRIGHT with gpt4_reason queries. The results are consistent with the findings in Section 3.4, showing that non-CoT Direct rerankers consistently outperform their reasoning counterparts. Table 7 further presents the performance of Direct rerankers on BRIGHT with gpt4_reason queries. Among pointwise models, Direct-Point-4B achieves the best score of 33.3, followed by Direct-Point-8B with 32.0. For listwise models, both Direct-List-4B and Direct-List-8B obtain 35.3, outperforming reasoning-enhanced rerankers and providing further evidence that explicit reasoning does not lead to better reranking performance.

## C  PROMPTS FOR RERANKING

### C.1  PROMPT FOR POINTWISE RERANKING

In the pointwise setting, the reranker judges each query–passage pair independently. The non-reasoning version (**Direct-Point**) directly outputs a binary decision, as shown in Figure 5. The reasoning version (**Reason-Point**) additionally generates a rationale enclosed within `<think>` tags before giving the final decision, as shown in Figure 4.

### C.2  PROMPT FOR LISTWISE RERANKING

In the listwise setting, the reranker considers the entire candidate set and outputs a ranked order of passages. The non-reasoning version (**Direct-List**) directly produces the final ranking sequence, as shown in Figure 7. The reasoning version (**Reason-List**) first generates a reasoning trace in `<think>` tags and then outputs the final ranking within `<answer>` tags, as shown in Figure 6.

Table 6: Performance of Direct-Point, Reason-Point, Direct-List, and Reason-List under different training strategies on BRIGHT (gpt4_query).

| Model | Training | StackExchange | | | | | | | Coding | | Theorem-based | | | Avg. |
|---|---|---|---|---|---|---|---|---|---|---|---|---|---|---|
| | | Bio. | Earth. | Econ. | Psy. | Rob. | Stack. | Sus. | Leet. | Pony | AoPS | TheoQ. | TheoT. | |
| **Pointwise** | | | | | | | | | | | | | | |
| Reason-Point-4B | SFT | 46.2 | 49.8 | 26.1 | 37.0 | 22.0 | 26.2 | 28.4 | 24.7 | 23.9 | 6.5 | 33.5 | 20.2 | 28.7 |
| Direct-Point-4B | SFT | 50.8 | 54.7 | 31.2 | 41.7 | 25.8 | 30.6 | 32.1 | 29.1 | 28.8 | 7.8 | 38.5 | 25.0 | **33.0** |
| Reason-Point-8B | SFT | 42.0 | 44.5 | 27.0 | 36.4 | 23.1 | 28.7 | 30.2 | 28.1 | 18.9 | 7.5 | 24.5 | 19.5 | 27.5 |
| Direct-Point-8B | SFT | 46.9 | 49.2 | 31.5 | 41.7 | 27.0 | 33.6 | 35.3 | 34.6 | 22.5 | 9.7 | 28.8 | 22.9 | 32.0 |
| **Listwise** | | | | | | | | | | | | | | |
| Reason-List-4B | SFT | 50.9 | 45.5 | 27.5 | 38.0 | 28.5 | 28.8 | 34.0 | 20.1 | 21.5 | 6.9 | 24.0 | 31.0 | 29.7 |
| Direct-List-4B | SFT | 53.5 | 48.3 | 29.8 | 39.9 | 31.2 | 30.5 | 36.9 | 23.0 | 25.0 | 8.2 | 26.3 | 34.2 | 32.2 |
| Reason-List-8B | SFT | 51.4 | 47.0 | 29.5 | 40.1 | 29.0 | 29.9 | 35.5 | 21.9 | 26.2 | 7.7 | 27.5 | 32.0 | 31.5 |
| Direct-List-8B | SFT | 54.0 | 49.6 | 30.1 | 42.0 | 31.6 | 31.8 | 37.1 | 24.1 | 28.0 | 8.4 | 28.1 | 35.0 | 33.3 |
| Reason-List-4B | SFT+GRPO | 55.1 | 49.0 | 28.7 | 42.7 | 31.1 | 33.1 | 36.6 | 21.8 | 23.8 | 7.8 | 27.7 | 37.2 | 32.9 |
| Direct-List-4B | SFT+GRPO | 58.4 | 51.8 | 31.3 | 41.6 | 34.4 | 33.4 | 41.0 | 25.2 | 27.9 | 9.8 | 29.5 | 39.2 | **35.3** |
| Reason-List-8B | SFT+GRPO | 54.5 | 49.4 | 30.8 | 44.4 | 29.9 | 32.1 | 38.7 | 22.6 | 28.0 | 8.9 | 31.4 | 36.0 | 33.9 |
| Direct-List-8B | SFT+GRPO | 56.1 | 52.2 | 30.3 | 44.8 | 32.5 | 36.4 | 38.8 | 25.0 | 30.9 | 8.5 | 29.6 | 38.9 | 35.3 |

Table 7: Performance of different rerankers on BRIGHT datasets (gpt4_query).

| Model | Training | StackExchange | | | | | | | Coding | | Theorem-based | | | Avg. |
|---|---|---|---|---|---|---|---|---|---|---|---|---|---|---|
| | | Bio. | Earth. | Econ. | Psy. | Rob. | Stack. | Sus. | Leet. | Pony | AoPS | TheoQ. | TheoT. | |
| **Pointwise** | | | | | | | | | | | | | | |
| BM25 | / | 53.6 | 53.6 | 24.3 | 38.6 | 18.8 | 22.7 | 25.9 | 17.7 | 19.3 | 3.9 | 20.2 | 18.9 | 26.5 |
| MonoT5-3B | SFT | 16.0 | 24.0 | 17.7 | 19.5 | 8.0 | 10.5 | 19.5 | 17.2 | 29.2 | 7.1 | 20.3 | 12.0 | 16.8 |
| RankLLaMA-7B | SFT | 17.5 | 15.5 | 13.1 | 13.6 | 17.9 | 6.9 | 16.9 | 8.4 | 46.8 | 2.2 | 4.5 | 3.5 | 13.9 |
| RankLLaMA-13B | SFT | 21.6 | 19.1 | 16.3 | 14.0 | 15.7 | 7.7 | 18.5 | 8.8 | 31.1 | 1.7 | 4.4 | 4.9 | 13.7 |
| Rank1-7B | SFT | 48.8 | 36.7 | 20.8 | 35.0 | 22.0 | 18.7 | 36.2 | 12.7 | 31.2 | 6.3 | 23.7 | 37.8 | 27.5 |
| Rank1-14B | SFT | 49.3 | 37.7 | 22.6 | 35.2 | 22.5 | 20.8 | 33.6 | 17.7 | 33.2 | 8.4 | 22.5 | 41.4 | 28.7 |
| Rank1-32B | SFT | 49.7 | 35.8 | 22.0 | 37.5 | 22.5 | 21.7 | 35.0 | 18.8 | 32.5 | 10.8 | 22.9 | 43.7 | 29.4 |
| Direct-Point-4B | SFT | 50.8 | 54.7 | 31.2 | 41.7 | 25.8 | 30.6 | 32.1 | 29.1 | 28.8 | 7.8 | 38.5 | 25.0 | **33.0** |
| Direct-Point-8B | SFT | 46.9 | 49.2 | 31.5 | 41.7 | 27.0 | 33.6 | 35.3 | 34.6 | 22.5 | 9.7 | 28.8 | 22.9 | 32.0 |
| **Listwise** | | | | | | | | | | | | | | |
| RankZephyr-7B | SFT | 44.1 | 31.0 | 17.9 | 28.4 | 17.5 | 27.0 | 21.6 | 18.9 | 17.8 | 2.7 | 15.9 | 12.7 | 21.3 |
| Rank-K | SFT | 50.4 | 46.2 | 30.6 | 46.7 | 32.4 | 33.0 | 41.2 | 24.0 | 32.2 | 7.6 | 28.3 | 26.6 | 33.3 |
| ReasonRank-7B | SFT+GRPO | 56.4 | 51.2 | 28.4 | 43.4 | 31.0 | 31.9 | 39.1 | 23.0 | 7.6 | 8.1 | 29.9 | 39.1 | 32.4 |
| Direct-List-4B | SFT+GRPO | 58.4 | 51.8 | 31.3 | 41.6 | 34.4 | 33.4 | 41.0 | 25.2 | 27.9 | 9.8 | 29.5 | 39.2 | **35.3** |
| Direct-List-8B | SFT+GRPO | 56.1 | 52.2 | 30.3 | 44.8 | 32.5 | 36.4 | 38.8 | 25.0 | 30.9 | 8.5 | 29.6 | 38.9 | **35.3** |

# D EXAMPLES FOR RANKING OUTPUT

## D.1 EXAMPLES FOR POINTWISE RANKING

Direct-Point rerankers perform a single forward pass per input and predict a binary answer token (`true`/`false`); we report the logits at the answer position, so the model does not explicitly print the answer string. In contrast, Reason-Point first generates a natural-language rationale and then computes the logits at the answer position, thereby outputting the complete reasoning text. For clarity, Figure 8 presents an output instance, where we additionally decode the answer-position logits into the corresponding answer token.

## D.2 EXAMPLES FOR LISTWISE RANKING

Figures 6 and 10 illustrate a concrete example of the listwise reranking setting. Figure 6 shows the input prompt template provided to the model, where the query and candidate passages are listed, and the model is instructed to return a complete ranking. Figure 10 presents the corresponding outputs

```
<|im_start|>system
Determine if the following passage is relevant
to the query. Answer only with 'true' or 'false'.
<|im_end|>
<|im_start|>user
Query: {}
Passage: {}
<|im_end|>
<|im_start|>assistant
```

Figure 4: Prompt template for pointwise relevance judgement.

```
<|im_start|>system
Determine if the following passage is relevant
to the query. Answer only with 'true' or 'false'.
<|im_end|>
<|im_start|>user
Query: {}
Passage: {}
<|im_end|>
<|im_start|>assistant

<think> </think>
```

Figure 5: Prompt template for pointwise relevance judgement (non-reasoning version). The `<think>` tag is kept empty to maintain consistency with reasoning prompts.

from different variants of our models. Non-reasoning models (Direct-List) directly produce the ranked sequence enclosed in `<answer>` tags, while reasoning models (Reason-List) first generate an explicit rationale enclosed in `<think>` tags before outputting the final ranking. This comparison highlights how reasoning influences the ranking process, providing intermediate explanations at the cost of increased verbosity.

## E    REASONING ASSUMPTION IN RANKING AND VERIFICATION OF REASONING TRACE QUALITY

This section presents quantitative and qualitative analyses to verify that the reasoning-based models generate valid and meaningful reasoning traces. The goal is to ensure that the observed performance trends are not attributable to poorly trained or defective reasoning behavior.

### E.1    REASONING ASSUMPTION IN RANKING

Reranking with explicit reasoning implicitly relies on a core assumption: *the generated chain-of-thought should provide a logically valid, coherent, and decision-supportive explanation that reflects the model's underlying relevance judgment.*

This assumption has been adopted—either explicitly or implicitly—in recent reasoning-enhanced ranking systems, where the reasoning trace is expected to: (1) extract or reference query–document evidence relevant to the final decision; (2) maintain internal logical consistency; and (3) articulate the decision boundary in a manner aligned with the teacher or supervision signal.

Under this assumption, if explicit reasoning improves the quality of the relevance decision, we would expect the reasoning-based reranker to produce *accurate, coherent, and faithful reasoning traces*, which subsequently translate into better ranking performance.

```
You are RankLLM, an intelligent assistant that can
rank passages based on their relevance to the query.
Given a query and a passage list, you first think
about the reasoning process in the mind and then
provide the answer (i.e., the reranked passage list).

The reasoning process and answer are enclosed
within <think> </think> and <answer> </answer>
tags, respectively, i.e.,
<think> reasoning process here </think>
<answer> answer here </answer>.

I will provide you with {num} passages, each
indicated by a numerical identifier [].
Rank the passages based on their relevance to
the search query:

[1]: {{passage_1}}
[2]: {{passage_2}}
(more passages)...

Search Query: {{query}}.
Rank the {num} passages above based on their
relevance to the search query.

All passages should be included and listed using
identifiers, in descending order of relevance.
The format of the answer should be [] > [],
e.g., [2] > [1].
```

Figure 6: Prompt template for listwise reranking with explicit reasoning. The model produces a reasoning trace within <think> tags and a final ranking within <answer> tags.

Therefore, our analyses proceed from the following question: *Do reasoning-based rerankers indeed learn to generate high-quality reasoning traces that support the relevance decision?*

To answer this, we conduct both quantitative and qualitative verification of reasoning trace quality in the following sections.

### E.2 QUANTITATIVE VERIFICATION

To assess whether the reasoning model has successfully learned to produce teacher-level decisions, we evaluate its binary true/false classification accuracy without applying any ranking procedure. The reported results (Table 3) compare the teacher model (DEEPSEEK-R1), the reasoning-based student (REASON-POINT-8B), and the direct baseline (DIRECT-POINT-8B) across both in-domain (MS MARCO) and out-of-domain (BRIGHT–Biology) settings.

The REASON-POINT-8B model achieves an average accuracy of **67.5**, closely matching the teacher model (**68.6**). This demonstrates that the student has successfully learned the teacher's decision behavior, indicating that the performance degradation observed in ranking cannot be attributed to failed reasoning acquisition.

### E.3 QUALITATIVE VERIFICATION

We further assess the validity of generated reasoning traces through a targeted qualitative study. A total of 50 query–document pairs were randomly sampled from MS MARCO and BRIGHT. For each pair, reasoning chains produced by DEEPSEEK-R1 and REASON-POINT-8B were independently evaluated by three annotators using a three-dimensional rubric:

```
You are RankLLM, an intelligent assistant that can
rank passages based on their relevance to the query.
Given a query and a passage list, directly provide
the reranked passage list without generating any
reasoning process.

I will provide you with {num} passages, each
indicated by a numerical identifier [].
Rank the passages based on their relevance to
the search query:

[1]: {{passage_1}}
[2]: {{passage_2}}
(more passages)...

Search Query: {{query}}.
Rank the {num} passages above based on their
relevance to the search query.

All passages should be included and listed using
identifiers, in descending order of relevance.
The format of the answer should be [] > [],
e.g., [2] > [1].
```

Figure 7: Prompt template for listwise reranking without explicit reasoning (non-reasoning version). The `<think>` tag is kept empty to maintain format consistency with reasoning prompts.

- **Correctness**: logical appropriateness of the conclusion,
- **Coherence**: internal consistency of reasoning steps,
- **Faithfulness**: structural alignment with the teacher's reasoning.

Majority-vote results are summarized in Table 8.

| Evaluation Dimension | DeepSeek-R1 | Reason-Point-8B | Agreement (%) |
|---|---|---|---|
| Correctness | 88% | 86% | 83% |
| Coherence | 92% | 88% | 81% |
| Faithfulness | – | 83% | – |

Table 8: Qualitative evaluation of reasoning trace quality.

The results show that REASON-POINT-8B produces reasoning traces that are largely correct (86%), coherent (88%), and structurally aligned with the teacher (83%). These findings confirm that the model generates valid and well-formed reasoning, and that the observed ranking behavior is not attributable to defective or low-quality reasoning traces.

## F  THE USE OF LARGE LANGUAGE MODELS (LLMS)

In the preparation of this manuscript, we employed `GPT-5` to assist with text refinement. Specifically, the model was used to improve the clarity, readability, and overall presentation of the paper by correcting grammatical errors, smoothing sentence structures, and enhancing stylistic consistency. Importantly, all core research ideas, experimental designs, and results were conceived and validated by the authors; the role of the LLM was limited to linguistic polishing. This ensured that the scientific content remained entirely authored by the researchers, while benefiting from improved academic writing quality.

```
<|im_start|>system
Determine if the following passage is relevant
to the query. Answer only with 'true' or 'false'.
<|im_end|>
<|im_start|>user
Query: Have governments ever defaulted on their
domestic public debt or treasury bonds due to
hyperinflation or political crises?

Passage: Argentina experienced a unique default
when it suspended all debt payments before a legal
default was declared, with 60% of the debt held
by Argentines. Rating agencies still classified it
as a default. Russia defaulted on its domestic debt
(GKOs) in 1998, which became one of the most
significant financial crises of that decade.
<|im_end|>
<|im_start|>assistant

<think>
Okay, let's tackle this query. The user is asking if
governments have ever defaulted on their public domestic
debt or treasury bonds, specifically due to hyperinflation
or political issues. The passage mentions Argentina and
Russia. Argentina suspended all payments, considered a
default by agencies, and Russia defaulted on GKOs in 1998.
Both are clear cases of default on domestic debt. While the
passage does not explicitly state hyperinflation, these are
examples that confirm the query.
...
</think>
true
```

Figure 8: Example of a Reason-Pointwise output with explicit reasoning in the `<think>` block before producing the binary decision.

```
Search Query: where is a elephant habitat.

Passages:
[1] Elephant Habitat. Elephants make home in a variety of habitats
    including tropical and subtropical zones...
[2] Elephant Natural Habitat. Elephants are able to survive in a
    variety of different locations...
[3] This is because the elephant is well known to being intelligent
    and capable of experiencing...
...
[20] Elephant Habitat. Elephants make home in a variety of habitats
    including tropical and subtropical zones...

Rank the 20 passages above based on their relevance to the query.
All the passages should be included and listed using identifiers, in
    descending order.
The format of the answer should be [] > [], e.g., [2] > [1].
```

Figure 9: Example listwise prompt for the query *"where is a elephant habitat"* with 20 candidate passages.

---

**Direct-List-8B_SFT**:

```
<think>
</think>
<answer> [8] > [19] > [11] > ... > [14] > [3] > [4] </answer>
```

**Reason-List-8B_SFT**:

```
<think>
Okay, let's tackle this ranking problem. The user wants to know where
    elephants live.
I need to focus on passages that explicitly describe habitats (
    savannas, forests, deserts, etc.).
Passage [1] lists tropical zones, dry forests, savannas, semi-deserts,
     rainforests.
[2] and [5] mention grasslands, deserts, forests, swamps.
[8] mentions Africa and South Asia. [10] and [11] describe
    adaptability.
Passages [4], [6], [7], [9], [19] repeat similar information.
[13] talks about different species. [15] and [17] focus on
    conservation.
[18] and [20] similar to [1] and [16].
Less direct: [3], [14], [12].
So top: [1], [2], [5], [8], [10], [11], [19], [13], [16], [20].
</think>
<answer> [1] > [2] > [5] > ... > [12] > [3] > [14] </answer>
```

**Direct-List-8B_GRPO**:

```
<think>
</think>
<answer> [8] > [19] > [1] > ... > [6] > [13] > [14] </answer>
```

**Reason-List-8B_GRPO**:

```
<think>
Okay, let's tackle this query about where elephants live. The user
    wants to know their habitats.
Passages that directly mention savannas, forests, deserts, and
    rainforests are key.
Looking at [1], [2], [5], [8], [10], [11], [16], [19], [20] all list
    habitats.
[8] and [19] mention Africa and South Asia, which is important.
Others talk about captivity or human impact, less relevant.
So the top ones are those listing habitats explicitly.
</think>
<answer> [1] > [2] > [5] > ... > [17] > [18] > [14] </answer>
```

Figure 10: Examples of model outputs on the elephant habitat ranking task. Non-reasoning models (NL) directly output rankings, while reasoning models (RL) generate intermediate rationales within <think> tags before the final answer.

