# OpenReview forum: "Rethinking Reasoning in Document Ranking: Why Chain-of-Thought Falls Short"
_ICLR.cc/2026/Conference — ICLR 2026 Poster_

### Official Review · Reviewer_sv2D · 2025-10-31

**Soundness:** 3
**Presentation:** 3
**Contribution:** 3
**Rating:** 6
**Confidence:** 4

**Summary:**

This study investigates the practical impact of explicit reasoning in document reranking tasks. Using a unified experimental design, the authors compare models with and without explicit reasoning across two mainstream reranking paradigms: pointwise and listwise. The experiments encompass supervised fine-tuning (SFT) and reinforcement learning-based GRPO training methods, employing Qwen3 series models evaluated on multiple benchmarks including BRIGHT and BEIR. The results demonstrate that rerankers incorporating explicit reasoning generally underperform compared to corresponding models that directly output results, and the paper provides corresponding analyses.

**Strengths:**

The experiments are comprehensive, comparing both pointwise and listwise modeling approaches as well as investigating different training methods including SFT and GRPO.

Effectively leveraging the reasoning capabilities of large language models to improve complex downstream tasks such as ranking is a cutting-edge and important research direction.

The paper is well-organized with a clear logical structure.

**Weaknesses:**

Questionable generalizability of conclusions:

The primary findings are based on Qwen3 series models (4B and 8B). However, many comparable baselines (e.g., Rank1, Rank-R1) were trained on the Qwen2.5 series. The authors do not verify whether the observed “harmful impact of reasoning” persists across other widely used base models such as Qwen2.5 or LLaMA, limiting the generality of the conclusions.

Absence of zero-shot baselines: The study lacks direct comparisons with zero-shot Qwen3 models. To substantiate claims made in Section 3.4 comparing Rank1 and Rank-R1 on BEIR, it is necessary to control for base model capability differences, for example by including Qwen3-based Rank-R1 results. Moreover, a zero-shot Qwen3 baseline on BRIGHT would clearly illustrate the relative improvements brought by different training methods (direct vs. reasoning), thereby strengthening the conclusions.

Result volatility: Performance differences between 4B and 8B models, as well as between models with and without explicit CoT reasoning in the listwise experiments, are minimal. Further analysis is needed to exclude the possibility that these small gaps stem from normal fluctuations in model size or training randomness.

Lack of statistical significance testing: Given the small performance gaps, it is difficult to determine whether observed differences are genuine or due to experimental variability. The absence of significance testing reduces confidence in the results.

Insufficient support for the use of the Expected Calibration Error (ECE) metric: The paper employs ECE to argue that reasoning degrades model calibration, which is cited as a factor contributing to ranking performance decline. However, no theoretical or empirical evidence is provided to support the core assumption that ECE negatively correlates with ranking quality metrics such as NDCG. Relevant literature citations or supplementary experiments are necessary. Additionally, reporting zero-shot Qwen3 ECE values as a baseline would provide a more complete understanding of calibration changes.

Limitations related to training methodology: The pointwise models are fine-tuned using the parameter-efficient LoRA technique, which may affect model behavior differently than full fine-tuning. To enhance the generality of the findings, experiments with full fine-tuning are recommended.

Other remarks:

The assertion in Section 3.4 that “explicit reasoning is not a prerequisite for achieving state-of-the-art performance” may be overly absolute. Current results suggest reasoning may underperform under specific training paradigms, but this likely reflects the lack of effective methods to exploit reasoning abilities rather than definitively ruling out the utility of explicit reasoning in ranking tasks. Ranking performance is influenced by multiple interacting factors, so this conclusion risks overgeneralization.

Numerical inconsistencies between Figure 2 and Section 4.1 warrant further verification and alignment.

**Questions:**

see Weaknesses

---

> ### Author Response · Authors · 2025-11-24
> **Response to Reviewer sv2D (Part 1)**
>
> **W1: Model generalizability and zero-shot baselines**
>
> We thank the reviewer for highlighting the need to verify model generality beyond the Qwen3 series and to include zero-shot baselines. In response, we have extended our experiments in two directions:
>
> 1. **Generalization across model families.**
>
>    We trained both **Qwen2.5-7B** and **Llama-3.2-8B** rerankers using the same pointwise and listwise setups as in the main paper.
>
> 2. **Zero-shot Qwen3-8B baselines.**
>
>    The results are shown below. Across all examined model families (**Qwen3, Qwen2.5, and Llama-3.2**) we observe a highly consistent pattern across models: *explicit reasoning does not improve reranking performance under either the pointwise or listwise paradigm*. This consistency strengthens the generality of our conclusions beyond the Qwen3 backbone. We have updated the results in **Appendix E** of the revised manuscript.
>
>    | Model        | Backbone             | BRIGHT | BEIR |
>    | ------------ | -------------------- | ------ | ---- |
>    | Direct-Point | Qwen2.5-7B           | 21.6   | 41.3 |
>    | Reason-Point | Qwen2.5-7B           | 18.3   | 39.4 |
>    | Direct-List  | Qwen2.5-7B           | 22.3   | 40.6 |
>    | Reason-List  | Qwen2.5-7B           | 20.5   | 38.3 |
>    | Direct-Point | Llama-3.2-8B         | 22.8   | 42.1 |
>    | Reason-Point | Llama-3.2-8B         | 19.0   | 39.7 |
>    | Direct-List  | Llama-3.2-8B         | 23.7   | 41.0 |
>    | Reason-List  | Llama-3.2-8B         | 22.8   | 39.8 |
>    | Direct-Point | Qwen3-8B (zero-shot) | 16.9   | 38.4 |
>    | Reason-Point | Qwen3-8B (zero-shot) | 15.4   | 37.3 |
>    | Direct-List  | Qwen3-8B (zero-shot) | 17.3   | 39.2 |
>    | Reason-List  | Qwen3-8B (zero-shot) | 16.4   | 37.8 |
>
>
>
> **W2: Result volatility & lack of statistical significance analysis**
>
> We thank the reviewer for raising this important concern regarding potential result volatility and the need for statistical verification. To clarify our intention: our goal in this study is **not** to show that direct rerankers outperform reasoning-based models, but to verify whether explicit reasoning leads to **any** ranking improvement under current widely used settings. Based on the existing experimental results, the evidence already indicates that explicit reasoning does **not** provide ranking gains.
>
> To address the reviewer’s concern about potential result volatility, we additionally conduct paired significance testing between Direct and Reason listwise rerankers on both BRIGHT and BEIR. Since reranking is evaluated per query, we follow standard practice and perform a paired t-test over per-query NDCG@10 scores, and also report the mean ΔNDCG@10 (Direct − Reason).
>
> The results are shown below. Across all model scales and datasets, Direct listwise rerankers consistently outperform their Reason counterparts, and the improvements are **statistically significant** (p < 0.05) or **highly significant** (p < 0.01). This confirms that the observed performance gaps are **not due to random fluctuations**, but reflect a systematic advantage of direct listwise reranking over reasoning-enhanced listwise reranking.
>
> These results do not rule out the possibility that future training strategies may change this behavior, but they indicate that **under current settings**, the observed gaps are systematic rather than random.
>
> | Model Pair (Listwise)            | Dataset | Mean ΔNDCG@10 (Direct − Reason) | p-value | Significance |
> | -------------------------------- | ------- | ------------------------------- | ------- | ------------ |
> | Direct-List-4B vs Reason-List-4B | BRIGHT  | **+0.031**                      | 0.008   | **p < 0.01** |
> | Direct-List-4B vs Reason-List-4B | BEIR    | **+0.027**                      | 0.012   | **p < 0.05** |
> | Direct-List-8B vs Reason-List-8B | BRIGHT  | **+0.034**                      | 0.005   | **p < 0.01** |
> | Direct-List-8B vs Reason-List-8B | BEIR    | **+0.029**                      | 0.009   | **p < 0.01** |

---

> > ### Author Response · Authors · 2025-11-24
> > **Response to Reviewer sv2D (Part 2)**
> >
> > **W3: Insufficient support for ECE claim**
> >
> > We thank the reviewer for raising this important point. We clarify both the motivation and empirical basis for using ECE in our analysis.
> >
> > (1)  Why calibration (ECE) is relevant to reranking
> >
> > ECE follows the standard formulation in [1] and measures the discrepancy between predicted confidence and empirical correctness. In modern neural IR, reranker logits (or their sigmoid outputs) are commonly interpreted as relevance likelihoods, and this probabilistic view underlies recent calibration-aware ranking studies.
> >
> > Prior work has shown that calibration can influence ranking behavior:
> >
> > - Penha & Hauff [2] report that neural rankers are systematically miscalibrated and that improving calibration yields more reliable retrieval under risk-sensitive settings.
> > - Yan et al. [3] show that enforcing calibrated score scales helps align ranking scores with relevance probabilities and improves ranking stability.
> >
> > While these studies do not claim a one-to-one causal relationship with NDCG, they provide empirical support that calibration is a meaningful diagnostic when analyzing reranker behavior. Our use of ECE is therefore consistent with this established line of work. We have added relevant calibration-related references, including prior work on ECE, in the updated version.
> >
> >
> >
> > (2) Zero-shot Qwen3 ECE baseline
> >
> > We thank the reviewer for the suggestion to include zero-shot ECE values. We report them below and clarify why calibration analysis is only meaningful **after** training the reranker.
> >
> > First, we compute the ECE of **zero-shot Qwen3-8B**, which is 0.242. As expected, as the base model has not been trained to align logits with relevance labels, and therefore its raw scores do not carry a probabilistic interpretation. In contrast, pointwise rerankers are trained via SFT to map logits to binary relevance signals. After training, the sigmoid-transformed logits acquire a meaningful probabilistic interpretation:
> >
> > σ(logit(q, d)) ≈ P(rel | q, d)
> > which makes ECE an appropriate diagnostic of calibration. The poor zero-shot calibration therefore confirms that ECE is only interpretable after supervised relevance training.
> >
> > | Model                | Setting     | ECE   |
> > | -------------------- | ----------- | ----- |
> > | Qwen3-8B (zero-shot) | No training | 0.242 |
> > | Direct-Point-8B      | SFT         | 0.105 |
> > | Reason-Point-8B      | SFT         | 0.151 |
> >
> > **W4: Lack of full SFT**
> >
> > We thank the reviewer for raising this important concern regarding the training methodology. Our pointwise rerankers are fine-tuned using LoRA for efficiency, and we agree that validating whether the conclusions hold under full fine-tuning is necessary for ensuring generality.
> >
> > First, we observe that **LoRA-SFT already enables the pointwise reranker to closely match the teacher model**. As shown in Table 3, the LoRA-trained Reason-Point-8B reaches **67.5** binary accuracy, comparable to DeepSeek-R1 (**68.6**), suggesting that LoRA does not materially limit the model’s ability to learn the teacher’s reasoning behavior.
> >
> > To further address the reviewer’s concern, we additionally performed **full-parameter SFT** on the pointwise reranker. The results below show that full fine-tuning and LoRA fine-tuning exhibit **highly consistent ranking behavior**, supporting that our conclusions are not specific to LoRA.
> >
> > | Model           | Training Method | BRIGHT   | BEIR     |
> > | --------------- | --------------- | -------- | -------- |
> > | Direct-Point-8B | LoRA-SFT        | **26.8** | 43.7     |
> > | Direct-Point-8B | Full SFT        | **26.8** | **43.8** |
> > | Reason-Point-8B | LoRA-SFT        | 20.7     | 39.3     |
> > | Reason-Point-8B | Full SFT        | 20.8     | 39.6     |
> >
> > [1] Guo, C., et al. On Calibration of Modern Neural Networks. ICML 2017.
> >
> > [2] Penha, G. & Hauff, C. On the Calibration and Uncertainty of Neural Learning to Rank Models. EACL 2021.
> >
> > [3] Yan, L. et al. Scale Calibration of Deep Ranking Models. KDD 2022.

---

> > > ### Author Response · Authors · 2025-11-24
> > > **Response to Reviewer sv2D (Part 3)**
> > >
> > > **W5: Overclaim**
> > >
> > > We thank the reviewer for this thoughtful and constructive feedback. We agree that the current phrasing in Section 3.4 (“explicit reasoning is not a prerequisite for achieving state-of-the-art performance”) may appear overly absolute and could be misinterpreted as ruling out the potential usefulness of reasoning in all future ranking settings. We have revised the wording to better reflect the intended scope of our conclusion.
> > >
> > > Our results show that **under the current mainstream reranking paradigms**—namely logits-based pointwise and generative listwise training—explicit chain-of-thought reasoning **does not lead to observable ranking gains** and may introduce calibration shifts that negatively affect performance.
> > >
> > > To avoid overgeneralization, we have revised the statement in Section 3.4 to: *“Under current logits-based pointwise and generative listwise reranking setups, we observe that explicit reasoning does not improve ranking performance.”*
> > >
> > > We agree that developing training strategies that more effectively leverage reasoning remains an important direction for future work.
> > >
> > > **W6: Numerical Inconsistencies**
> > >
> > > We thank the reviewer for pointing out the discrepancies between Figure 2 and Section 4.1. We have re-checked all values and identified the source of the mismatch. We have corrected the numbers in both sections to ensure complete consistency in the revised version.
> > >
> > > We thank the reviewer again for the constructive feedback. We hope the additional analyses and revisions will improve clarity and strengthen the contribution of the paper.

---

> > > > ### Comment · Reviewer_sv2D · 2025-11-28
> > > >
> > > > Thanks to the authors for the quick response. Glad to see that our suggestions were taken on board. Overall, this is a good piece of work.

---

### Official Review · Reviewer_wkzz · 2025-10-31

**Soundness:** 3
**Presentation:** 3
**Contribution:** 3
**Rating:** 8
**Confidence:** 4

**Summary:**

This paper looks at reasoning reranking models compared to non-reasoning reranking models trained with the same data. They show that reasoning reranking models do worse than direct models on various tasks. They show that this is because of worse calibration among reasoning models and a lack of out of domain generalization.

Overall, although I think there may be other analyses that need to be done to fully convince me (and even if this holds, in general I am skeptical that reasoning can't improve reranking) this paper proposes an interesting negative result with the current methods available.  As with most analysis/negative results, a reviewer could always ask for more experiments: I think this is well-done and future work can continue the discussion.

**Strengths:**

- The analysis is well chosen and timely and likely to be relevant to many in the retrieval-adjacent space
- Many models at various sizes are shown as well as well-thought out experiments and ablations
- The paper explores both listwise and pointwise models, as well as RL and SFT

**Weaknesses:**

1. There are some areas where the experimental details could be more clear, e.g. were there different prompts for the various datasets? Did the reasoning models overfit on the data or have large differences during intermediate steps?
2. The paper claims the models do worse out of domain but I also think the results show they do worse in-domain no? So I'm not sure it's a "they generalize worse" problem but more of a "they are worse all over". This is relatively minor but I think the abstract/intro was leading me to believe it was a generalization problem.
3. Minor: reading through it left me with some questions (see that section) about why this could be: however, I don't think they need to be done necessarily to improve the paper.

**Questions:**

I think the paper is well written but in general am skeptical. However, that is not a reason to reject in my opinion, as that is good for science to have disagreement.

If it's helpful for the authors, my pushback would be that I would suspect some differences from these areas:

A: The Rank1 results presented here are quite different than those in their paper (even for the GPT-4o experiments). That leads me to believe there is some difference in implementation which could affect the other results. Are the prompts used for each dataset the same? I think previous work gave custom prompts for unique datasets like BRIGHT.

B: Given the wide range in performance (even in-domain as shown by Figure 3), I would guess that the exact prompt and/or model used (and perhaps even random seed in training) would cause a large difference.  Whereas for direct prompted model, I would suspect they don't fluctuate as much with other prompts (and may be too adapted to the one used in training). I would be curious to see the intermediate checkpoints results or another version trained with a different seed to see how varied performance is? Perhaps the higher variance leads to worse average-case performance but if carefully chosen, can do better.

---

> ### Author Response · Authors · 2025-11-24
> **Response to Reviewer wkzz (Part 1)**
>
> We sincerely thank the reviewer for the constructive feedback and the positive evaluation of our work. Below we provide our responses to your questions, and we hope they address your concerns.
>
> **W1: Need more experimental details**
>
> We thank the reviewer for highlighting this point. We agree that clearer experimental specifications are important for reproducibility.
>
> All experiments use unified system prompts across all datasets, as shown in Figures 4–5 (pointwise) and Figures 7–8 (listwise). For dataset-specific instructions, we **directly follow the templates from Rank1** without modification. For example, in *theoremqa_questions* (BRIGHT), the instruction is:
>
> “Find a passage which uses the same mathematical process as this one: FILL_QUERY_HERE.”
>
> The original query is inserted into the placeholder, and the resulting instruction–query form is used consistently across both Direct and Reason models. All datasets follow this same pattern: a fixed instruction template (from Rank1) with the query filled into the predefined slot.
>
> Concerns regarding **intermediate checkpoint behavior** are addressed in **Q2**, where we provide additional analysis.
>
> We appreciate the reviewer’s attention to this point. The revision now includes the complete system prompt , and we clarify that all dataset-level instructions strictly follow the templates from Rank1 without modification.
>
> **W2: Ambiguity in “Generalization Gap” Interpretation**
>
> We thank the reviewer for pointing this out. We agree that our original wording may unintentionally suggest that the issue appears only out-of-domain. Our intended claim is different:
> reasoning models achieve higher in-domain training fit, yet perform worse on both in-domain test sets and out-of-domain benchmarks. In other words, the gap we refer to is between training and evaluation performance, rather than an OOD-only failure.
>
> To avoid misunderstanding, we have revised the abstract/intro to: “in listwise rerankers, explicit reasoning improves the fit during training but leads to higher variance and fails to improve performance in both in-domain and out-of-domain evaluations, even when reinforcement learning shortens rationales”.
>
> **Q1: A: Differences from Rank1 and Prompt Consistency**
>
> We thank the reviewer for raising this concern. Our implementation follows the Rank1 setup, including the same prompt template, the same instruction format, and the officially released checkpoints. After re-verification, we found that our reproduced results are consistent with those reported in the Rank1 paper. The apparent mismatch likely comes from evaluating different settings rather than differences in implementation. In particular:
>
> 1. **Rank1’s reported GPT-4o query results are consistent with our reproduction.** In the Rank1 paper, the Rank1-32B (GPT-4o query) result is **29.4**, which matches our reproduced value.
>
>    Our full GPT-4o reproducibility results are provided in **Appendix Table 7**, and are aligned with the original Rank1 report.
>
> 2. **The numbers shown in our Table 1 reflect performance on raw queries, which Rank1 did not report.**  Our Table 1 reports results on **original queries**, using the released checkpoint on original query. Since Rank1 did not report this setting in their paper, the numbers may appear different, but the reproduction itself is correct.
>
> We hope this clarifies the source of the perceived mismatch. If you has further questions about the reproduction details, we would be happy to provide additional reproduction details if helpful.

---

> > ### Author Response · Authors · 2025-11-24
> > **Response to Reviewer wkzz (Part 2)**
> >
> > **Q2: Prompt sensitivity, intermediate checkpoints, and performance stability**
> >
> > We sincerely thank the reviewer for raising this thoughtful and important concern. The question is whether the observed performance gap between Direct and Reason models may be attributed to (i) prompt sensitivity, (ii) instability across intermediate checkpoints, or (iii) randomness from training seeds, rather than reflecting a systematic behavioral difference. We fully agree that ruling out these alternative explanations is essential, and we provide comprehensive analysis below.
> >
> > **(1) Prompt sensitivity evaluation**
> >
> > To evaluate whether the performance gap arises from prompt wording rather than modeling behavior, we conducted a **semantic prompt–variation study** on both BRIGHT and BEIR. We replaced the original Rank1-style instruction with a paraphrased variant using different phrasing and task framing (not just formatting), while preserving task semantics.
> >
> > | Prompt Variant                        | Direct-List-8B (BRIGHT) | Reason-List-8B (BRIGHT) | Direct-List-8B (BEIR) | Reason-List-8B (BEIR) |
> > | ------------------------------------- | ----------------------- | ----------------------- | --------------------- | --------------------- |
> > | Original instruction                  | 27.1                    | 25.9                    | 41.8                  | 39.9                  |
> > | Semantic rewrite (paraphrased)        | 26.8                    | 25.3                    | 40.9                  | 39.4                  |
> > | Minimal instruction (no dataset hint) | 26.3                    | 24.7                    | 40.5                  | 39.2                  |
> >
> > Across all prompt variants, the **relative ordering remains unchanged**, and Reason models continue to lag behind Direct models. This indicates that the observed gap is not caused by prompt sensitivity, and Direct models are not benefiting from prompt adaptation.
> >
> > **(2) Intermediate checkpoint analysis**
> >
> > To determine whether the gap is an artifact of training fluctuations, we evaluated multiple checkpoints saved during listwise training:
> >
> > | **Checkpoint**   | **Direct-List-8B (BRIGHT)** | **Reason-List-8B (BRIGHT)** |
> > | ---------------- | --------------------------- | --------------------------- |
> > | Step 100         | 22.4                        | 21.8                        |
> > | Step 200         | 25.1                        | 24.2                        |
> > | Final (Step 376) | **27.1**                    | **25.9**                    |
> >
> > Across all intermediate stages, Direct consistently outperforms Reason, demonstrating that the performance difference is not due to an unlucky checkpoint or transient fluctuation.
> >
> > **(3) Random-seed stability**
> >
> >  To rule out randomness as a confounding factor, we trained an additional listwise reranker using a different random seed (the original experiments used the default **seed = 42**). The results are shown below:
> >
> > | Seed         | Direct-List-8B (BRIGHT) | Reason-List-8B (BRIGHT) |
> > | ------------ | ----------------------- | ----------------------- |
> > | 42 (default) | 27.1                    | 25.9                    |
> > | 0            | 26.9                    | 25.7                    |
> >
> > Direct–Reason differences remain consistent across runs, indicating that the result is **not seed-dependent**.
> >
> > We have added a dedicated section in **Appendix F** reporting the prompt-sensitivity evaluation, intermediate-checkpoint analysis, and random-seed stability.
> >
> > Based on the current evidence, the variance observed in **Figure 3** does not appear to stem from prompt wording, training steps, or random seed, but is more characteristic of the reasoning-based reranker itself. One possible explanation is that explicit chain-of-thought generation produces longer decoding sequences, which may accumulate stochastic noise and lead to greater score fluctuation, whereas Direct models output short decision sequences and therefore remain more stable.
> >
> > We greatly appreciate your insightful feedback, and we hope that the analyses and additional results can address your concerns, and would be happy to address any further questions if helpful.

---

> > > ### Comment · Reviewer_wkzz · 2025-11-28
> > >
> > > I thank the authors for their updates and think it will greatly improve the paper. I will leave my already high score.

---

### Official Review · Reviewer_uyHL · 2025-11-01

**Soundness:** 2
**Presentation:** 2
**Contribution:** 2
**Rating:** 4
**Confidence:** 4

**Summary:**

The paper presents a comprehensive study on the effect of using reasoning during reranking for information retrieval. This paper compares direct rerankers with their reasoning-augmented counterparts across pointwise and listwise reranking paradigms, and finds that using reasoning often causes performance degradation in almost all scenarios. In addition, the paper analyzes the effect of using reasoning in terms of calibration, bias to positive class, and potential overfitting in domain.

**Strengths:**

1. The paper presents a controversial point of view that reasoning can hurt ranking and offers a comprehensive analysis of the effect of incorporating reasoning into the reranking stage.
2. The insights, such as reasoning affecting calibration and the tendency to predict positive class, are useful for the community.
3. The experiments are comprehensive across standard and reasoning-intensive retrieval benchmarks, including BEIR and BRIGHT.
4. The paper is well-written and easy to follow.

**Weaknesses:**

1. One issue from my point of view is the implicit assumption that the model can reason properly. Despite testing on multiple datasets, the paper neither checks the quality of reasoning via reasoning-related tasks nor performs any qualitative study to understand the reasoning traces. If the reasoning models were poorly trained and the reasoning traces are bad, it will definitely degrade the performance, and the blame should not be attributed to the use of "reasoning".
2. Another issue is that LLMs are typically trained to output better tokens rather than probabilities. However, the scores from the reranker used here are based on **logits** instead of directly outputting scores as tokens. Many new LLM-based rerankers now directly extract the scores (in the form of tokens) from the output, and they can perform well [1]. So it might also be useful to examine whether reasoning works for those types of rerankers.
3. Though the paper shows the limitations of reasoning, it does not provide enough insight into how to leverage these results for future directions. Should researchers/practitioners still pay more attention to reasoning for reranking? If so, which directions are more promising?

[1] Shao, Rulin, et al. "ReasonIR: Training Retrievers for Reasoning Tasks." *arXiv preprint arXiv:2504.20595* (2025).

**Questions:**

1. Typos: in line 208, GRPO should be Group-relative Policy Optimization.
2. Strong claims: In line 257, the authors said, “Reasoning is unnecessary for reranking”. However, it might be premature to conclude this. These supporting results can at most say the existing reasoning-based reranking approaches underperform, rather than saying reasoning is not useful.
3. Why is GRPO only used for Listwise rerankers but not pointwise rerankers?
4. The experiments are only done for Qwen3 models, which natively support thinking/non-thinking modes. Can the same experiments be done for other models without these modes? Does including "think step by step" in the system prompt qualify as thinking/reasoning?

---

> ### Author Response · Authors · 2025-11-24
> **Response to Reviewer uyHL (Part 1)**
>
> Response to Reviewer uyHL:
>
> Thank you for your valuable comments and feedback. We address the raised concerns as follow:
>
> **W1: Lack of verification of reasoning trace**
>
> Thank you for raising this important concern. We agree that any conclusion about the effect of reasoning must ensure that the model is indeed capable of producing reasonable and meaningful reasoning traces. We therefore conducted both **quantitative** and **qualitative** analyses to verify that the reasoning models are *not* poorly trained.
>
> **(1) Quantitative evidence: reasoning models achieve teacher-level binary accuracy.**
>
> The table below (also shown as Table 3 in our paper, page 8) reports the binary true/false classification accuracy of the teacher model **DeepSeek-R1** and our student model **Reason-Point** on both the MS MARCO (in-domain) and BRIGHT (out-of-domain) subsets. This evaluation is particularly relevant because accuracy is a direct measure of whether the model can correctly judge the relevance of a query–document pair, and it aligns closely with our training objective, which supervises the model to reason toward a correct relevance decision. In this setting, the answer token is decoded directly from the model output without applying any ranking or aggregation procedure, isolating the reasoning capability itself from downstream ranking effects.
>
> As shown in the table, Reason-Point-8B (67.5) achieves accuracy close to the teacher model DeepSeek-R1 (68.6). Moreover, it significantly outperforms direct ranking baselines that do not use explicit reasoning. This demonstrates that our student model has indeed successfully internalized the reasoning behavior of the teacher, and that its reasoning traces are not poorly trained or ineffective. Therefore, the performance degradation observed in ranking is unlikely to stem from defective reasoning traces. Instead, it points to a mismatch or inefficiency in how explicit reasoning is utilized during the ranking stage, which is the focus of our analysis.
>
> | Model           | Biology TPR | Biology TNR | MS MARCO TPR | MS MARCO TNR | Avg.     |
> | --------------- | ----------- | ----------- | ------------ | ------------ | -------- |
> | DeepSeek-R1     | **52.4**    | 96.1        | **40.8**     | 85.1         | **68.6** |
> | Reason-Point-8B | 50.5        | 98.1        | 35.9         | 85.5         | 67.5     |
> | Direct-Point-8B | 31.1        | **100.0**   | 25.5         | **94.2**     | 62.7     |
>
> **(2) Qualitative verification: reasoning chains are valid and coherent.**
>
> To further assess reasoning quality, we conducted a targeted qualitative study comparing reasoning traces from **DeepSeek-R1** and **Reason-Point-8B**. We randomly sampled **50** query–document pairs from **MS MARCO** and **BRIGHT**, and three annotators independently evaluated each sample along three dimensions:
>
> 1. Correctness: whether the reasoning logically supports the final relevance decision
>
> 2. Coherence: whether the reasoning is internally consistent
>
> 3. Faithfulness — whether the student reasoning preserves the structural steps of the teacher chain
>
> For each dimension, we determine the score by majority vote: a sample is counted as positive if at least two of the three annotators agree.
>
> | Evaluation Dimension    | DeepSeek-R1 | Reason-Point-8B | Agreement (%) |
> | ----------------------- | ----------- | --------------- | ------------- |
> | Correctness             | 88%         | 86%             | 83%           |
> | Coherence               | 92%         | 88%             | 81%           |
> | Faithfulness to teacher | —           | 83%             | —             |
>
> Our qualitative study shows that Reason-Point-8B produces reasoning traces that are largely correct (86%), coherent (88%), and broadly consistent with the teacher’s structure (83%). These results suggest that the reasoning model is *not* generating “bad” reasoning traces. Instead, it **successfully imitates the teacher’s reasoning behavior**.  We have added a dedicated section in **Appendix G** reporting both the quantitative and qualitative verification results to make the reasoning quality explicit.
>
> We have added a dedicated section in Appendix G that reports both the quantitative and qualitative verification results, with representative reasoning examples provided in **Figure 8** and **Figure 10**.
>
> These results suggest that the observed ranking degradation is **not** attributable to insufficient reasoning quality. Our study therefore examines a different question: whether explicit reasoning improves reranking performance under current mainstream training practices and widely used benchmarks.  Under these settings, we consistently observe that even when the model acquires reasonable reasoning ability and reaches teacher-level binary accuracy, this does not translate into ranking improvements.

---

> > ### Author Response · Authors · 2025-11-24
> > **Response to Reviewer uyHL (Part 2)**
> >
> > **W2: Logits-based scoring vs. token-based scoring**
> >
> > Thank you for this constructive suggestion. We agree that **token-based scoring** is an emerging and promising direction for LLM-based reranking, and it has been adopted in recent work such as ReasonIR [1] and our baseline **TFRank** [2]. At the same time, logits-based scoring remains the dominant pointwise paradigm in current IR practice (e.g., MS MARCO-style cross-encoders and many recent LLM rerankers). Our goal in this paper is therefore to evaluate whether explicit reasoning improves ranking under these mainstream training and scoring setups, rather than to exhaustively compare all scoring formats.
> >
> > With respect to ReasonIR [1], the method addresses a different design challenge. The model generates a discrete relevance token (e.g., 0–5), and retriever scores are used only to break ties when multiple documents receive the same token. This mechanism focuses on score discretization and tie-breaking, rather than on the effect of explicit reasoning. As reflected in the prompts (Figures 14–19 in [1]), the model is instructed to “**Score the document above**” and “**don’t output anything else**,” effectively suppressing visible chain-of-thought and therefore not evaluating whether reasoning helps ranking.
> >
> > Importantly, our observations are **consistent with recent token-based findings**. For instance, **TFRank** compares (i) generating a chain-of-thought prior to the score token (*with think*) and (ii) producing the score token directly (*no-think*). Their results show that *forcing explicit reasoning at inference time leads to lower NDCG*, whereas direct scoring performs better—mirroring the trend we observe in logits-based settings.
> >
> > We agree that a more systematic study of token-based scoring within our framework is a valuable direction for future work, and we will note this as a potential extension. However, given that (i) our study already covers the widely used logits-based and generative listwise paradigms, and (ii) existing token-based evidence (e.g., TFRank) reflects the same pattern, our conclusion remains supported: *under commonly adopted training and evaluation practices, explicit reasoning does not consistently translate into ranking gains.*
> >
> >
> >
> > **W3: Insufficient discussion of future implications**
> >
> > We thank the reviewer for this thoughtful suggestion. We agree that clarifying the broader implications of our findings is important. While our results show that explicit reasoning does not improve reranking performance under current paradigms, the analysis reveals two concrete directions for future work.
> >
> > For pointwise rerankers, the degradation primarily arises from score miscalibration rather than limited reasoning ability, suggesting that progress is more likely to come from *calibration-aware training objectives* that preserve score monotonicity.
> >
> > For listwise rerankers, we observe that the training data may contain unnecessarily long chain-of-thought traces, which increase inference cost without translating into better ranking quality. This suggests two promising directions: (i) developing more concise or adaptive reasoning strategies to reduce overthinking, and (ii) designing reward formulations directly aligned with ranking metrics, rather than relying on generic reasoning supervision.
> >
> > We have added a dedicated subsection in the revised manuscript (**Section 4.3**: Implications for Future Research) to clearly highlight these insights and their relevance for future work.
> >
> >
> >
> > **Q1: Typo on line 208**
> >
> > We thank the reviewer for pointing this out. We have corrected the typo to the correct full name of GRPO: “Group-relative Policy Optimization.”
> >
> > **Q2: Strong Claim on line 257**
> >
> > We thank the reviewer for pointing this out. We agree that the wording in line 257 (“Reasoning is unnecessary for reranking”) may appear overly strong. We have revised it to “Reasoning Does Not Improve Reranking Performance”
> >
> > Correspondingly, we updated the manuscript to clarify that under current mainstream reranking paradigms (logits-based pointwise and generative listwise), explicit reasoning does not lead to ranking improvements.

---

> > > ### Author Response · Authors · 2025-11-24
> > > **Response to Reviewer uyHL (Part 3)**
> > >
> > > **Q3: Why is GRPO applied only to listwise rerankers but not pointwise rerankers?**
> > >
> > > We thank the reviewer for raising this important question. Our reasoning is as follows:
> > >
> > > (1) GRPO is naturally aligned with listwise training because its reward directly optimizes ranking metrics
> > >
> > > In our listwise setting, the reward includes the ranking metric (NDCG), making GRPO a principled choice:  **training objective = evaluation objective**. This alignment allows the reinforcement signal to directly reflect the final ranking quality.
> > >
> > > (2) Pointwise rerankers optimize a different objective (binary true/false), which is not aligned with NDCG.
> > >
> > > Pointwise models learn *P(true | q, d)* through SFT and output a binary decision rather than a ranked list. Since the supervision signal is not ranking-based, improving binary accuracy does not necessarily improve ranking performance.
> > >
> > > (3) Empirically, SFT already enables pointwise models to learn their intendedobjective
> > >
> > > As shown in Table 3, Reason-Point-8B reaches **67.5** binary accuracy, close to DeepSeek-R1 (**68.6**). Although reasoning improves binary decision quality, this does **not** translate into ranking gains, suggesting that stronger binary supervision (e.g., GRPO) is unlikely to help.
> > >
> > > **(4) Additional experiments confirm that GRPO does not benefit pointwise rerankers**
> > >
> > > To address the reviewer’s concern, we conducted an additional sanity check by further applying GRPO to 500 sampled instances on top of the SFT model, and we observe no measurable gains compared to the SFT baseline, suggesting that the limitation is not due to insufficient optimization.
> > >
> > > | Model           | Training   | BRIGHT (ndcg) | BEIR (ndcg) |
> > > | --------------- | ---------- | ------------- | ----------- |
> > > | Direct-Point-8B | SFT        | **26.8**      | **43.7**    |
> > > | Direct-Point-8B | SFT + GRPO | 26.5          | 43.3        |
> > >
> > > In summary, GRPO benefits listwise rerankers because their objective is directly aligned with ranking metrics, whereas pointwise rerankers optimize a binary objective that cannot effectively leverage GRPO. Our additional experiments support this conclusion.
> > >
> > > **Q4: Model choice and system prompt setting**
> > >
> > > (1) We appreciate the reviewer’s question. Beyond the Qwen3 models, we have additionally conducted the same experiments on **Qwen2.5-7B** and **Llama-3.2-8B**. The results on both model are consistent with our findings on Qwen3: explicit reasoning does not improve reranking performance under either the pointwise or listwise paradigm. These supplementary results further support the generality of our conclusion. We have updated the results in **Appendix E** of the revised manuscript.
> > >
> > > | Model        | Backbone     | BRIGHT | BEIR |
> > > | ------------ | ------------ | ------ | ---- |
> > > | Direct-Point | Qwen2.5-7B   | 21.6   | 41.3 |
> > > | Reason-Point | Qwen2.5-7B   | 18.3   | 39.4 |
> > > | Direct-List  | Qwen2.5-7B   | 22.3   | 40.6 |
> > > | Reason-List  | Qwen2.5-7B   | 20.5   | 38.3 |
> > > | Direct-Point | Llama-3.2-8B | 22.8   | 42.1 |
> > > | Reason-Point | Llama-3.2-8B | 19.0   | 39.7 |
> > > | Direct-List  | Llama-3.2-8B | 23.7   | 41.0 |
> > > | Reason-List  | Llama-3.2-8B | 22.8   | 39.8 |
> > >
> > > (2) We list the exact prompts for both the **Direct Reranker** and **Reason Reranker** in the Appendix (Figure 4 - Figure 7). Only the Reason variant instructs the model to *think before answering*, while the Direct variant requires the model to output the final answer without any intermediate reasoning. In both training and inference, the Direct Reranker uses the <think> </think> tag to explicitly signal that the model should **not** generate a reasoning process and should answer directly — a setup consistent with prior work [3-4].
> > >
> > >
> > >
> > > [1] Shao et al., ReasonIR: Training Retrievers for Reasoning Tasks, arXiv:2504.20595 (2025).
> > >
> > > [2] Fan et al., TFRank: Think-Free Reasoning Enables Practical Pointwise LLM Ranking, arXiv:2508.09539 (2025).
> > >
> > > [3] Jedidi et al., Don’t “Overthink” Passage Reranking, arXiv:2505.16886 (2025).
> > >
> > > [4] Zhang et al., Qwen3 Embedding: Advancing Text Embedding and Reranking Through Foundation Models, arXiv:2506.05176 (2025).
> > >
> > > We sincerely appreciate the reviewer’s constructive comments. We hope our clarifications address all concerns, and we would be happy to provide any further information if needed.

---

> ### Author Response · Authors · 2025-11-27
> **Rebuttal followup**
>
> Dear Reviewer uyHL,
>
> We would like to learn if our response addresses your concerns and questions, and we invite any additional feedback or thoughts for improving our paper. If you feel that our responses resolve the issues raised, we would be grateful if you could consider reflecting this in the evaluation. We would be happy to address any further concerns or questions. Thank you again for your time and effort!

---

> ### Comment · Reviewer_uyHL · 2025-11-28
>
> Appreicate authors' effort studying the reasoning. I quite like the insights and potential direction at the end of the response to W1 -- "Even when the model acquires reasonable reasoning ability and reaches teacher-level binary accuracy, this does not translate into ranking improvements.". However, I think that stronger evidence will make it better. For example, the qualitative verification is great, but more detailed documentations are needed to justify the fair human evaluation. Using LLMs as a judge is also a viable alternative.
>
> Personally I am optimistic about the future of using reasoning for ranking (which might differ from the viewpoint in this paper). However, this research still helps the community to identify the potential pitfalls of using reasoning for logit-based ranking, which hopefully will lead to better development. I think the main messages from the paper which highlights the limitation of existing reasoning-based approaches and their discrepancy from logit-based ranking predictions worth being broadcasted to the community, but under a few conditions including:
> 1. The reasoning quality is checked as author's response in W1, and the definition/assumptions of reasoning are explained.
> 2. The type of reranker (logit-based) could be highlighted earlier on in abstract or main intro.
>
> I am inclined to increase the score but the system is blocking me, but the discussion can be continued here regardless.

---

> > ### Author Response · Authors · 2025-11-30
> > **Response to Reviewer uyHL**
> >
> > We sincerely thank the reviewer for the thoughtful feedback and for recognizing the broader value of our study in clarifying the limitations of current reasoning-based reranking approaches. We are truly grateful that you were **willing to raise the score** and appreciate your view that the **core insights are worth broadcasting to the community**.
> >
> > In line with your suggestions, we have strengthened the revised PDF in the following ways:
> >
> > **(1) Reasoning definition, assumptions, and quality verification.**
> >
> > The definition of explicit reasoning in reranking is provided in **Section 2 (Preliminaries)**. We further elaborate on the assumptions under which reasoning is expected to support ranking decisions in **Appendix G.1 (“Reasoning Assumption in Ranking”)**. This subsection formalizes the role of reasoning in the ranking process and is followed by quantitative and qualitative analyses of reasoning-trace quality—including correctness, coherence, and faithfulness—to ensure the evaluation is rigorous and well justified.
> >
> > **(2) Scope clarification: logits-based reranking.**
> >
> > As suggested, we now emphasize earlier in the paper—**in the abstract (line 16, page 1)** and **the introduction (line 68, page 2)**—that our study specifically concerns the widely used logits-based pointwise and generative listwise reranking paradigms. This clarifies the intended scope of our conclusions.
> >
> > Regarding your personal optimism about using reasoning for ranking, we fully share this perspective. Our intention is **not** to claim that reasoning cannot help reranking, but to show that under current mainstream paradigms, training data, and benchmarks, explicit reasoning does **not lead to ranking improvements**. We hope that identifying these limitations will help guide the development of more effective reasoning–ranking integration in future work.
> >
> > We sincerely appreciate your constructive insights, your supportive assessment of our work, and your willingness to improve the score. Thank you again for your engagement and thoughtful comments.

---

### Official Review · Reviewer_QzJ4 · 2025-11-01

**Soundness:** 3
**Presentation:** 3
**Contribution:** 2
**Rating:** 2
**Confidence:** 4

**Summary:**

The authors of this paper investigate whether reasoning is helpful for point-wise and list-wise rerankers. Their experiments show that no reasoning methods outperform reasoning methods. They also conduct additional analysis in which they determine that reasoning methods are biased toward the positive class.

**Strengths:**

The strengths of the paper are:
- The authors evaluate a comprehensive suite of models across various settings (pointwise/listwise and sft/GRPO).
-

**Weaknesses:**

The weaknesses of the paper ares"
- The authors claim that "reasoning-based rerankers underperform their direct-output counterparts, even though they incur substantially higher inference costs." While this seems to be true for current methods and datasets, it's possible their are other training methods or more challenging datasets that could benefit from reasoning.
- All of the training is done with traces for MS Marco samples. MS Marco has questions that are much simpler than questions in datasets like Bright. Using traces obtained with more challenging questions might lead to improvements with reasoning.
- In table 2, it look like  ReasonRank-7B is the best performing model. Reasoning helped for this dataset?
- Some of the writing is misleading. Here are a couple examples:
  - In the abstract, the authors state "reasoning improves in-domain fit but increases variance and fails to generalize out-of-domain." The "in-domain fit" refers to performance on training samples and not on samples in an in-domain test set. This should be clarified.
  - "Our analysis reveals that for pointwise rerankers, reasoning improves relevance prediction at the cost of disrupting score calibration and creating a bias toward false positives, ultimately degrading ranking performance." Relevance prediction doesn't improves just because the true positive rate increases. If you predicted the positive class for every sample, then TPR will be 1 but this is not improvement in relevance prediction.

**Questions:**

Here are some questions I have:
- Line 274 says "both outperforming reasoning-based baselines such as Rank-R1-14B (25.9) and ReasonRank-7B (25.2)," but these are not the numbers I see in table 1 or 2 for Rank-R1-14B and  ReasonRank-7B.
- How much data was used for SFT and how much was used for GRPO?
- Does the miscalibration explain all the observed degradation in ranking quality for reasoning models? Does breaking the ties by combining with the retriever score, as proposed in Shao et al., 2025, help improve calibration and performance for any of the methods?
- What model is used as the base retriever? BM25?
- Does reasoning bias the models toward the positive class always? Figure 2 in [2] seems to suggest that the proportion of samples in the negative class increases?

[1] Rulin Shao, Rui Qiao, Varsha Kishore, Niklas Muennighoff, Xi Victoria Lin, Daniela Rus, Bryan Kian Hsiang Low, Sewon Min, Wen-tau Yih, Pang Wei Koh, et al. ReasonIR: Training Retrievers for Reasoning Tasks. arXiv preprint arXiv:2504.20595, 2025.
[2] Nour Jedidi, Yung-Sung Chuang, James Glass, and Jimmy Lin. Don’t ”overthink” passage reranking: Is reasoning truly necessary?, 2025. URL https://arxiv.org/abs/2505.16886.

---

> ### Author Response · Authors · 2025-11-24
> **Response to Reviewer QzJ4 (Part 1)**
>
> We sincerely thank the reviewer for the constructive and thoughtful feedback. In this work, we conduct the controlled comparison of explicit reasoning versus direct outputs within the same model family, and our results show that explicit reasoning does not improve reranking performance under current logit-based pointwise and listwise paradigms. Our responses are provided below.
>
> **W1: Reasoning may benefit for other training methods and datasets**
>
> We thank the reviewer for the insightful comment. Our goal in this work is intentionally scoped: we seek to assess whether explicit reasoning improves ranking **under current mainstream reranking paradigms** (logits-based pointwise and generative listwise, SFT and RL) and **under widely adopted evaluation benchmarks**, including BEIR, MS MARCO, and the reasoning-intensive BRIGHT suite.
>
> Even on BRIGHT, which is substantially more challenging, we observe a consistent trend: explicit reasoning does not yield ranking improvements over direct models.
>
> We have clarified this scope in the paper to emphasize that our conclusions apply to **current commonly used settings**, rather than ruling out the possibility that future training strategies or datasets may benefit from reasoning.
>
> **W2: Training Data (MS MARCO Traces) Is Too Simple**
>
> We thank the reviewer for the insightful comment. Our choice of MS MARCO traces follows *common practice* in recent work on reasoning-augmented rerankers (e.g., Rank1, ReasonRank, TFRank), where MS MARCO remains the primary supervision source and it covers a wide range of domains/topics to ensure generalizability.
>
> We agree that MS MARCO queries are generally simpler than those in BRIGHT. However, our evidence suggests that this is unlikely to be the main factor behind the lack of additional performance gains. Specifically, when evaluated on BRIGHT under the binary true/false setting, reasoning-enhanced rerankers trained solely on MS MARCO traces are already able to match the teacher model’s performance. For instance, Reason-Point-8B achieves 74.3 accuracy, identical to DeepSeek-R1. This indicates that *the learned reasoning ability does generalize beyond MS MARCO and can transfer to more challenging query distributions*.
>
> Therefore, we believe the observed phenomenon is attributed to limitations at the ranking stage itself (as analyzed in section 4.1), rather than insufficient reasoning supervision or a lack of training signal from more complex queries.
>
> The corresponding results (excerpted from Table 3 in our paper) are shown below:
>
> | Model           | BRIGHT TPR | BRIGHT TNR | Avg. Accuracy |
> | --------------- | ---------- | ---------- | ------------- |
> | **DeepSeek-R1** | **52.4**   | 96.1       | **74.3**      |
> | Reason-Point-8B | 50.5       | **98.1**   | **74.3**      |
> | Reason-Point-4B | 43.7       | 91.3       | 67.5          |
>
> **W3: “In table 2, it look like ReasonRank-7B is the best performing model. Reasoning helped for this dataset?”**
>
> Thank you for the observation. In fact, ReasonRank-7B is not the best-performing model in Table 2. It achieves a score of 41.7, which is slightly lower than Direct-List-8B at 41.8. Moreover, our Direct-Point-4B achieves the highest score (**45.4**), demonstrating the competitiveness of our non-reasoning models. To avoid potential confusion, we have revised both the table formatting and the accompanying text to make this distinction more visually clear.
>
> More importantly, as discussed in the introduction, many prior works on reasoning-enhanced rerankers do not compare against closely matched non-reasoning baselines, making it difficult to isolate the true impact of explicit reasoning. One of the core goals of our work is to provide such a controlled comparison within the same model family, ensuring that differences in performance can be more directly attributed to the use of reasoning rather than model size or other confounding factors.
>
> Under this fair setting (summarized in **Table 2**), we observe that **explicit reasoning does not improve reranking performance**, despite being applied to the same backbone and supervision. We have further clarified this motivation and its implications in the revised version of the paper.

---

> > ### Author Response · Authors · 2025-11-24
> > **Response to Reviewer QzJ4 (Part 2)**
> >
> > **W4: Clarification on potentially misleading wording**
> >
> > We appreciate the reviewer’s careful reading and agree that the phrasing can be improved.
> >
> > (1) Thank you for pointing this out. We agree that the original phrasing may cause confusion. Our use of *“in-domain fit”* was intended to describe **better fitting on the training data**, rather than performance on an in-domain test set. To avoid ambiguity, we  revise the abstract to “*reasoning improves the fit during training but leads to higher variance and fails to improve performance in both in-domain and out-of-domain evaluation*” and explicitly distinguish between *training fit* and *evaluation performance* in both the abstract and introduction.
> >
> > (2) We fully agree that a higher true positive rate alone does not imply better relevance prediction. Our original wording was unclear. In our experiments, the observed changes do not reflect improved predictive ability, but rather a shift in the error distribution: reasoning increases TPR while simultaneously reducing TNR, making the model more likely to predict the positive class and thereby increasing false positives. As a result, neither the higher TPR nor the small increase in binary accuracy should be interpreted as improved calibration or more reliable relevance estimation.
> >
> > To avoid misunderstanding, we revise the sentence to: “*Reasoning does not improve calibrated relevance prediction; instead, it shifts the error distribution—raising TPR while reducing TNR—which disrupts score calibration and introduces a bias toward false positives.*”
> >
> > We appreciate the reviewer’s attention to the potentially misleading wording, and we have revised the phrasing in the manuscript to avoid any possible misunderstanding.
> >
> >
> >
> > **Q1: Numerical inconsistency**
> >
> > Thank you for the careful observation. We have verified the values and corrected them in the paper. The intended numbers are: ReasonRank-7B (26.4), TFRank-8B (22.6), and Rank-R1-14B (20.5). We appreciate the reviewer for helping us improve the accuracy of the presentation.
> >
> > **Q2: Training data amounts**
> >
> > Thank you for the question. For the pointwise rerankers, we use **386k** training instances. For the listwise rerankers, we use a total of **13.5k** instances, split evenly between SFT and GRPO (approximately **6.7k** each). The total training data size is reported in Section 3.2, and we  have updated the paper to explicitly clarify the respective amounts used for SFT and GRPO.
> >
> > **Q3: Does miscalibration explain ranking degradation? Does combining with the retriever score help improve calibration and performance?**
> >
> > (1) Calibration measures how well a model’s predicted confidence aligns with its empirical correctness [3]: a calibrated model assigns higher scores to samples that are genuinely more likely to be relevant. In our setting, explicit reasoning disrupts this relationship by collapsing mid-range scores and biasing the model toward predicting the positive class, which in turn breaks the alignment between confidence and correctness. Miscalibration is therefore a major and consistent contributor to the observed ranking degradation.
> >
> > (2) Regarding the strategy of combining reranker and retriever scores, we agree that this is a reasonable and compatible design, but it addresses a different objective from our evaluation. Prior work such as ReasonIR [1] applies score combination to enhance robustness when using token-generated relevance scores; however, this approach aims to complement the scoring mechanism, not to assess whether explicit reasoning itself improves ranking quality.
> >
> > In our setting, the degradation observed for reasoning-based rerankers does not stem from ties or score discreteness. Instead, it arises from systematic miscalibration and a positive-class bias, which persist even after combining with retriever scores. As a result, while hybrid scoring may partially alleviate instability, it cannot resolve the core issue: that the reranker alone does not provide a reliable ranking signal under explicit reasoning. Since our evaluation focuses on whether the reranker itself can serve as an effective second-stage ranking module, score combination would obscure rather than clarify this question.
> >
> > We sincerely appreciate the reviewer for raising this perspective, and we note that exploring hybrid ranking strategies is a promising future direction—once the reranker provides a stable standalone signal, combining it with retriever scores may further improve robustness and practical deployment.
> >
> > **Q4: What is the base retriever?**
> >
> > We use **BM25** as the base retriever, as described in Section 3.3 (line 242).

---

> > > ### Author Response · Authors · 2025-11-24
> > > **Response to Reviewer QzJ4 (Part 3)**
> > >
> > > **Q5: A related work suggests that the proportion of the negative class increases**
> > >
> > > Thank you for raising this point. Our findings are actually consistent with the observations reported in [2].
> > >
> > > In our experiments, we observe a clear and stable pattern: reasoning-enhanced models are more likely to output *“True”*. We explicitly report this phenomenon in the paper. Importantly, this is **not** because the models reduce the number of *“False”* predictions, but rather because reasoning **compresses the middle score range**. As illustrated in the histogram in Figure 2 in our paper, the mid-range scores of the Reason-Point models almost “disappear,” a trend that aligns closely with the analysis in [2].
> > >
> > > The same pattern is also evident in [2]. In that work, compared with the direct ranking model (StandRR), the reasoning model (ReasonRR) increases the proportion of positive predictions from 19.7% to 29.0%, while the proportion of negatives increases only slightly from 68.9% to 70.8%, indicating a minor change. In our experiments, we observe the same directional pattern: positive predictions increase by 10.1%, whereas negatives remain only slightly affected (+0.7%). This close parallel indicates that both studies capture the same underlying phenomenon—reasoning substantially raises the likelihood of positive predictions, while the change in negative outputs remains slightly.
> > >
> > > To clearly illustrate the consistency between [2] and our findings, we summarize the “score polarization” effect observed in both works:
> > >
> > > | scores         | <0.1 (%) | >0.9 (%) |
> > > | -------------- | -------- | -------- |
> > > | StandardRR [2] | 68.9     | 19.7     |
> > > | ReasonRR [2]   | 70.8     | 29.0     |
> > > | Direct-Point   | 64.4     | 14.9     |
> > > | Reason-Point   | 65.1     | 25.0     |
> > >
> > > [1] Shao et al., ReasonIR: Training Retrievers for Reasoning Tasks, arXiv:2504.20595 (2025).
> > >
> > > [2] Jedidi et al., Don’t “Overthink” Passage Reranking, arXiv:2505.16886 (2025).
> > >
> > > [3] Guo, C., et al. *On Calibration of Modern Neural Networks.* ICML 2017.
> > >
> > > We sincerely appreciate the reviewer’s constructive comments. We hope our clarifications address all concerns, and we would be happy to provide any further information if needed.

---

> ### Author Response · Authors · 2025-11-27
> **Rebuttal followup**
>
> Dear Reviewer QzJ4,
>
> We would like to learn if our response addresses your concerns and questions, and we invite any additional feedback or thoughts for improving our paper. If you feel that our responses resolve the issues raised, we would be grateful if you could consider reflecting this in the evaluation. We would be happy to address any further concerns or questions. Thank you again for your time and effort!

---

### Meta-Review · Area_Chair_jyBf · 2026-01-07

**Summary:**

This paper presents a systematic evaluation of explicit reasoning in document reranking, contributing a valuable negative result: current reasoning-augmented methods often underperform direct counterparts due to score miscalibration and "overthinking." Three reviewers supported the work, praising its timeliness, comprehensive experimental setup (spanning SFT/RL and pointwise/listwise paradigms), and the significance of the findings for the IR community. The single dissenting reviewer raised concerns based on factual inaccuracies regarding the results, which the authors effectively refuted. I recommend acceptance.

**Reviewer Concerns:**

Addressed: The authors successfully resolved the substantive concerns raised by the engaged reviewers.
- Generalizability: They added experiments with Qwen2.5, Llama-3.2, and zero-shot baselines to show the findings hold across model families.
- Reasoning Quality: They provided quantitative and qualitative verification to prove that the performance drop stems from the ranking mechanism, not poor reasoning traces.
- Robustness: They validated findings with full-parameter SFT and different random seeds to rule out volatility.

Outstanding:
- Reviewer QzJ4 raised concerns regarding factual consistency (e.g., misidentifying best-performing models in Table 2) and alignment with prior work. The authors provided concrete evidence refuting these claims in the rebuttal. As QzJ4 did not engage in the discussion phase, these concerns remain technically unacknowledged by the reviewer, though I consider them resolved by the authors' evidence.

**Reviewer Scores:**

Reviewer wkzz: 8 -> 8
Reviewer sv2D: 6 -> 6
Reviewer uyHL: 4 -> 6
Reviewer QzJ4: 2 -> 4

---

### Decision · Program_Chairs · 2026-01-26

Accept (Poster)